# SkipW: Resource Adaptable RNN with Strict Upper Computational Limit

**Tsiry Mayet, Anne Lambert, Pascal Le Guyadec, Françoise Le Bolzer & François Schnitzler**
InterDigital Inc.
Cesson-Sévigné, France
`{firstname.lastname}@interdigital.com`

## Abstract

We introduce Skip-Window, a method to allow recurrent neural networks (RNNs) to trade off accuracy for computational cost during the analysis of a sequence. Similarly to existing approaches, Skip-Window extends existing RNN cells by adding a mechanism to encourage the model to process fewer inputs. Unlike existing approaches, Skip-Window is able to respect a strict computational budget, making this model more suitable for limited hardware like edge devices. We evaluate this approach on four datasets: a human activity recognition task, sequential MNIST, IMDB and adding task. Our results show that Skip-Window is often able to exceed the accuracy of existing approaches for a lower computational cost while strictly limiting said cost.

**Keywords:** Recurrent neural networks, Flexibility, Computational resources.

## 1 Introduction

Since Recurrent Neural Networks (RNN) have been introduced Williams et al. (1986), they have become one of the reference methods to process sequences. A typical architecture is the Long-Short-Term-Memory neural network (LSTM) which allowed improvement in natural language processing such as large-vocabulary speech recognition (Sak et al., 2014; Li & Wu, 2015). Used with CNNs they have also reached state of the art in automatic image captioning (Vinyals et al., 2015).

Deep learning models are now brought closer to the user rather than running in a distant cloud, helping to reduce latency, network congestion, and improving data security and privacy. However, smartphones and user devices impose additional constraints such as limited computation or energy. Handling these constraints has become an active research topic (Zhang et al., 2017; 2018; Howard et al., 2019; Wu et al., 2019; Cai et al., 2020). User devices can also host multiple processes running at the same time and starting or stopping abruptly, modifying the constraints affecting the processes. Few works have considered models that can be modified at run time to adapt to an evolving computational limit (Yu et al., 2019; Yu & Huang, 2019; Guerra et al., 2020; Jin et al., 2020). However, none of these focus on sequences and therefore none address the problem of adapting the model in the middle of a sequence.

In this context, this paper introduces Skip-Window (SkipW), a flexible recurrent neural network architecture: its computational cost can be dynamically adapted during a sequence analysis to meet real time constraints changes. The proposed architecture can be combined with any RNN cell and allows to strictly limit the computational resources used to avoid exceeding a given budget. Furthermore, empirical experiments on four data sets (Adding Task, MNIST, IMDB and HAR-2D-POSE) demonstrate that this subsampling architecture is interesting in itself. Skip-Window matches or exceed the accuracy of existing approaches for a given computational cost. In addition, measurements on specific processors highlight that SkipW produces real computational and energy savings.

## 2 Related work

Typically, RNNs maintain a "state", a vector of variables, over time. This state is supposed to accumulate relevant information and is updated recursively. Each input of the sequence is typically

a) processed by some deep layers and b) then combined with the previous state through some other deep layers to compute the new state. Hence, the RNN can be seen as a function taking a sequence of inputs $\boldsymbol{x} = (x_1, \ldots, x_T)$ and recursively computing a set of states $\boldsymbol{s} = (s_1, \ldots, s_T)$. Each state $s_t$ is computed from $s_{t-1}$ and $x_t$ by a cell $S$ of the RNN. As neural networks are increasingly run on limited hardware, recent research has focused on controlling their computational cost.

## 2.1 FLEXIBLE NEURAL NETWORKS

A few architectures have recently been designed to adapt the computational complexity of a Deep Neural Network (DNN) without reloading the whole model. This can be achieved by removing/adding neurons (Yu et al., 2019; Yu & Huang, 2019) or by modifying the quantization of the weights (Guerra et al., 2020; Jin et al., 2020). An efficient embedding of a mixture of Convolutional Neural Network (CNNs) also allows to add or remove several models at the same time, hence changing the computational cost (Ruiz & Verbeek, 2019).

### 2.1.1 THRRNN

For RNNs specifically, ThrRNN (Lambert et al., 2020) aims to control computation time by not processing some inputs. This is controlled by an update gate $u_t$. The tradeoff between the average accuracy and the average number of updates can be modified during inference by changing a single parameter $thr$. ThrRNN can wrap any RNN cell $S$:

$$u_t = f_{binarize}(\tilde{u}_t, thr) = \begin{cases} 0 & if \ \tilde{u}_t < thr \\ 1 & otherwise \end{cases} \tag{1}$$

$$\Delta \tilde{u}_t = \sigma(W s_t + b) \tag{2}$$

$$\tilde{u}_{t+1} = u_t \Delta \tilde{u}_t + (1 - u_t)(\tilde{u}_t + min(\Delta \tilde{u}_t, 1 - \tilde{u}_t)) \tag{3}$$

$$s_t = u_t S(s_{t-1}, x_t) + (1 - u_t)s_{t-1} \ . \tag{4}$$

When an input is processed, an update gate computes the quantity $\Delta \tilde{u}_t$ that determines how many inputs will be skipped. In practice the $\Delta \tilde{u}_t$ are accumulated in $\tilde{u}_t$ until $\tilde{u}_t \geq thr$.

## 2.2 RECURRENT NEURAL NETWORK WITH LOW COMPUTATIONAL COMPLEXITY

Several architectures have been proposed to limit or reduce the computational cost of RNNs, but this cost cannot be adapted at inference. A first class of architectures dynamically reduces computation based on the input. SkipRNN (Campos et al., 2018) predates and is similar to ThrRNN, except that the binarization function does not change. A similar mechanism has been proposed by Zhang et al. (2019). Other architectures directly select the next input to process (Yeung et al., 2016; Yu et al., 2017; Hansen et al., 2019; Song et al., 2018). Early exit has also been investigated by Dennis et al. (2019). Tao et al. (2019) also use $\mathbf{x}_t$ as input to an update gate. So do Seo et al. (2018); Jernite et al. (2017); Li et al. (2020). However, they do not skip any input but perform partial state updates.

A second class of architectures focuses on reducing the overal cost of the RNN. FastRNN is an RNN augmented with a residual connection with two extra scalar parameters and FastGRNN is an improved FastRNN: the residual connection is extended to a gate and RNN matrices are low rank, sparse and quantized (Kusupati et al., 2018). Other architectures reduce the RNN length. Chan et al. (2016) train an encoder to reduce the input length. Yeung et al. (2016); Shan et al. (2018); Chen et al. (2018) propose various mechanisms to summarize subsequences of windows of inputs.

## 2.3 RECURRENT NEURAL NETWORK WITH HIERARCHICAL-DEPENDENT COMPLEXITY

A class of architectures focuses on hierarchy level concept to reduce the complexity. These methods are mainly used in the context of multi-layer RNNs where each layer is supposed to model a different level in the hierarchy (e.g. for a corpus the levels could be documents, paragraphs, sentences, words, letters). These approaches are based on the fact that a hierarchical separation exists within a sequence of inputs, which might not always be the case.

In (Koutnik et al., 2014), the hidden state is partitioned into different modules, each module has its own clock period, meaning that they will be updated at different times. Skipping updates of part of

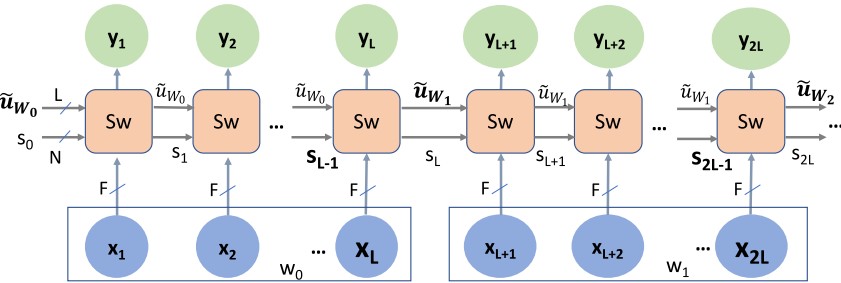

Figure 1: Skip-window computes update gates at the beginning of each window of inputs.

the hidden state decreases the computational cost. In (Koutnik et al., 2014), the update periods are chosen arbitrarily, for example using an exponential series.

For stacked-RNNs, Chung et al. (2017); Chang et al. (2017) conditionally update each layer based on a feature level criterion, or by dilating a skip connection. Layers close to the inputs would model lower feature levels and be updated more frequently. Further layers would model higher level features. In (Chung et al., 2017), a layer modeling sentences would be updated only when a word is processed entirely (in a model fed character by character), from the layer modeling words. Before the end of a word is reached, the state of the former layer is copied across input steps.

## 2.4 RELATIONSHIP TO OUR WORK

ThrRNN is the closest model to SkipW. Both are flexible RNNs and skip some inputs. However, ThrRNN optimizes computational cost on average over sequences. This induces two variabilities: a) inter-sequence variability: the model will not use the same number of updates for every sequence; and b) intra-sequence variability: the number of updates will not be uniform across time steps, updates may be concentrated in a certain part of the sequence. These two variabilities can cause the model to exceed its computational budget and, therefore, to either shut down or delay the output. SkipW does not have this problem as it strictly enforces a computational constraint over each window of inputs.

Other strategies for flexible models are not straightforward to apply to RNN. They require specialized training algorithms. They have never been applied to models processing inputs of an RNN or to make an RNN flexible and it is not clear how they would need to be modified. Furthermore, these models adapt between independent inputs whereas, for sequences, adaptation is necessary between time steps. RNN architectures with low complexity are orthogonal to our approach. They do not offer flexibility. They could be combined with and benefit from our approach. However, SkipRNN (which we are based on) and related methods have one big advantage over others: by skipping inputs, they also skip any modification of an input, such as processing by an expensive CNN for images. As SkipW makes decision over a window of inputs, it has some superficial similarity to methods summarizing windows or hierarchical RNNs. However, SkipW a) does not summarize windows and b) does not even look at these inputs before deciding what to skip.

## 3 SKIP-WINDOWS ARCHITECTURE

Skip-Windows (SkipW) is a wrapper for a RNN cell $S$. It uses a conditional computation mechanism to skip some updates. Rather than at each input $x_t$, update gates are computed at the beginning of windows of inputs, that is, every $L$ time steps (Figure 1). In other words, before any new L-size window of inputs, a L-size vector $\tilde{u}_W$ is computed. $\tilde{u}_W[i]$ can be seen as the importance of input $i$ in the window.

Then, the architecture includes a $select_K$ mechanism. This function takes as input the vector $\tilde{u}_W$ and outputs the vector $\tilde{u}_W^K$, setting $L - K$ elements to a value that ensures the associated inputs are not processed (0 in Figure 2). Therefore, it ensures that at most $K$ out of every $L$ inputs will be processed. In other words, it forces the RNN cell to skip $(L - K)$ out of every $L$ inputs. This

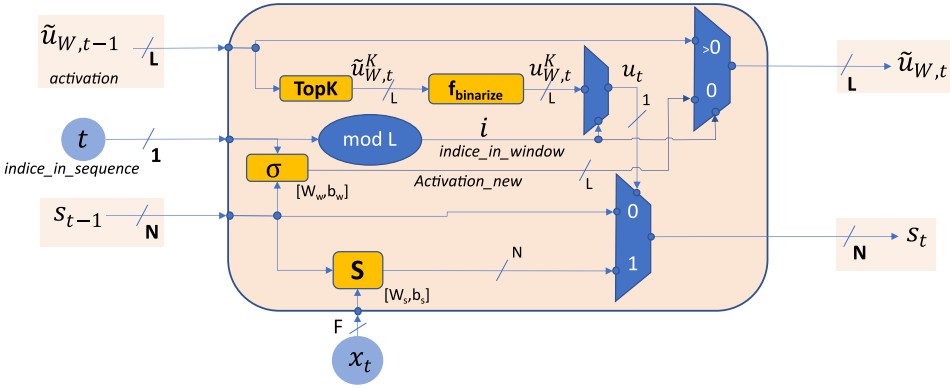

Figure 2: TopK-based Skip-Window cell implementation example.

ensures a strict upper bound on the computational cost of the model for a sequence and for each window, therefore alleviating the inter-sequence variability and intra-sequence variability issues.

Similarly to other works, the binary state update L-size vector, $u_W$, is then obtained by binarizing the remaining values as in equation 1. For example, by setting all values below a threshold to a value that ensures the associated inputs are not processed (0 in Figure 2).

An example of the Skip-Window cell implementation is represented in Figure 2. In this case, $select_K$ is implemented as a $top_K$ function. This enforces the strict constraint on the number of updates. The $top_K$ operation keeps unchanged the $K$ highest value in $\tilde{u}_{W,t}$, and sets the $(L - K)$ others to 0.

The corresponding architecture can be characterized as follows:

$$s_t = u_t \cdot S(s_{t-1}, x_t) + (1 - u_t) \cdot s_{t-1} \tag{5}$$

$$\tilde{u}_{W,t+1} = \gamma \cdot \sigma(W_w(s_{t-1}, t) + b_w) + (1 - \gamma) \cdot \tilde{u}_{W,t} \tag{6}$$

$$\gamma = \begin{cases} 1 & if \ i == 0 \\ 0 & otherwise \end{cases} \tag{7}$$

$$i = t \ mod \ L \tag{8}$$

$$\tilde{u}_{W,t}^K = top_K(\tilde{u}_{W,t}) \tag{9}$$

$$u_t = f_{binarize}(\tilde{u}_{W,t}^K[i], thr) = \begin{cases} 0 & if \ \tilde{u}_{W,t}^K[i] < thr \\ 1 & otherwise \end{cases} \tag{10}$$

where $W_w$ is a weight matrix of size $(N + 1) \times L$, $N$ is the number of hidden units as defined by the RNN cell S, $b_W$ is a L-vector bias, $\sigma$ is the sigmoid function and $mod$ is the modulo operation. Instead of a $top_K$ function for $select_K$, it is also possible to use a stochastic sampling mechanism that randomly selects (without replacement) $K$ out of $L$ elements of $\tilde{u}_W$ where the probability of selecting each element of index $i$ is proportional to $\tilde{u}_W[i]$. Some $select_K$ alternatives are discussed and evaluated in Appendix H.

Including the time step $t$ in equation 6 is also optional and can be replaced by a value ensuring the state is not static if no update is made in a window. For example, the number of inputs since the last update or the number of windows already processed.

**Training the model** The model is trained to minimize a two-part loss, similarly to Campos et al. (2018). The first term measures the accuracy of the task, and the second one penalizes inputs used:

$$L_{budget} = \lambda \sum_{t=1}^{T} u_t \ , \tag{11}$$

where $\lambda$ is the cost associated to the use of a single input. More experimental details are provided in Appendix B.

**Error gradients**  The model is differentiable except for the $f_{binarize}$ and $top_K$ function. To train the model using standard backpropagation the straight-through estimator is used as done in Campos et al. (2018) for $f_{binarize}$.

Other alternatives might involve reinforcement learning such as REINFORCE (Williams, 1992) or, in the case of the $top_K$ function, the usage of a differentiable $top_K$ as proposed in Xie et al. (2020). Early experiments using a differentiable $top_K$ (Xie et al., 2020) have shown worse results than the straight-through estimator. This suggests that constraining computation may be an interesting approximation for a $top_K$ operation.

**Adapting computational cost at inference**  During inference, adapting the tradeoff between model performance and computational cost can be done using two factors : the $K$ in equation 9 and the $thr$ in equation 10. These two parameters can be modified together or one at a time. Increasing/lowering $thr$ parameter in $[0, 1]$ encourages the model to process fewer/more inputs. Changing $K$ in $\{0 \ldots L\}$ forces the model to process at most $K/L$ of the window.

**Choice of the window size hyper-parameter**  By the nature of the model, the task can influence the choice of $L$. It can be hand tuned or computed using typical hyper-parameter search methods such as grid search. Choosing small $L$ allows the model to make update decisions for the near future only but offers less choice in operating points. Similarly a bigger $L$ requires the model to predict its update decisions for a bigger time span but offers more flexibility. At the extreme when $L = 1$, each window consists of a single input.

## 4  EXPERIMENTS

We empirically show that SkipW a) learns accuracy / computational tradeoffs on par with or, surprisingly, sometimes better than our baselines SkipRNN and ThrRNN on all data sets; b) upper bounds computational cost with gradual impact on accuracy except on sequential MNIST; c) enables changing the tradeoff in the middle of a sequence. To evaluate the computational benefit induced by skipping part of the input sequence we use the number of inputs processed by the model, as this measure is independent from the hardware and the framework used to implement the models. In addition, we also show how a reduction of the number of inputs processed translate into computational and energy savings on tiny devices.

We evaluate our approach on four data sets. **Human Activity Recognition (HAR)** (Ofli et al., 2013) consists of 2D-position sequences of 18 joints across a 32-frames videos recorded at 22Hz ($\approx$1.5 seconds). Each sequence is labeled by one of the 6 actions performed by 12 subjects filmed from 4 angles. Sequential **MNIST** (Lecun et al., 1998), the handwritten digits data set, consists of digital picture made of $28 \times 28$ pixels. Each image is flattened into a 784-sequence of one pixel. **Adding Task** Hochreiter & Schmidhuber (1997), consists of tuples-sequences (value, marker) $\in \mathbb{R} \times \{0, 1\}$, where the expected output is $\sum_i marker \times value$. **IMDB** (Maas et al., 2011) consists of movie reviews with a positive/negative annotation. The reviews were padded or cut to a length of 200 words in our experiment. For HAR, two stacked RNNs are used, but only one for Adding Task, Sequential MNIST and IMDB. More details can be found in Appendix A and experimental results on Adding Task, HAR and Sequential MNIST are further developed in Appendices D, F and E.

### 4.1  ACCURACY AND COMPUTATIONAL TRADEOFFS

In Figures 3 to 6, the baselines SkipRNN and ThrRNN are compared to SkipW with various values for $L$ and $K$ for all data sets. Small $L$ values may not offer good tradeoffs when varying $K$. See for example $L = 4$ in Figure 3. That's not surprising: often the number of inputs processed can be significantly reduced without affecting accuracy. For example, SkipRNN on HAR can process as few as 15% inputs without affecting accuracy. Using $L = 4$ and $K = 1$ limits complexity at 20% only. So, for such a small $L$, varying $K$ has almost no impact on average. Larger window sizes, on the other hand, offer more tradeoffs when changing $K$. For example, on HAR, for $L \in \{8, 16\}$, $K$ can take more values, and for small $K$ computational gains are traded off for accuracy. For example, for $L = 16$, reducing $K$ from 4 to 1 more than halves the number of inputs processed for a drop of 3.6% in accuracy. A similar behavior can be observed on Adding Task and IMDB. On MNIST (Figure 5), lowering $K$ trades off accuracy for computational cost at a worse ratio. This is the only

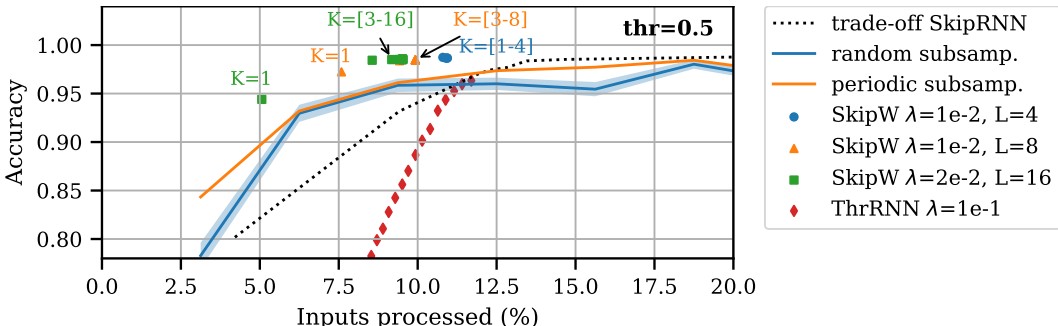

Figure 3: Impact of the parameter $K$ on accuracy/updates tradeoff for HAR. Comparison between ThrRNN (thr=0.5 to 1.0), SkipW($L$=4,8,16; $thr$=0.5, $K$=1 to $L$) and SkipRNN ($\lambda$=1e-3 to 3e-1). For random subsampling, the shaded area corresponds to 3 times the standard deviation on each side of the mean value (50 evaluations).

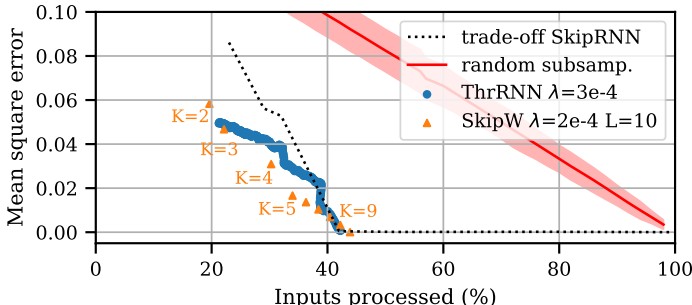

Figure 4: Impact of the parameter $K$ on mean square error and inputs processed for Adding Task. Comparison between ThrRNN ($\lambda$=3e-4, $thr$=0.5 to 1.0) and SkipW ($\lambda$=2e-4, $L$=10, $thr$=0.5, $K$=1 to $L$). The dotted line corresponds to the performance of individual SkipRNN models trained with different values of $\lambda$ (from $\lambda$=1e-5 to $\lambda$=5e-4) and the red line to a baseline randomly sampling inputs. The shaded area corresponds to 3 times the standard deviation around the mean (100 evaluations).

data set out of four where SkipW is unable to match the performance of SkipRNN while changing $K$. Even in this case, independent SkipW (with fixed $K = L$) provide comparable or better tradeoffs than SkipRNN (Appendix E). In addition, SkipW performs as well as ThrRNN on MNIST. Figures 3 to 4 also include a comparison to two naive subsampling strategies selecting a set number of inputs. *Random subsampling* samples inputs based on a uniform distribution. *Periodic subsampling* selects them at regular interval. On Adding Task, these methods are equivalent. Both naive strategies are worse than SkipW on both data sets.

SkipW enforces a $K/L$ upper bound on the number of inputs processed in each window. Even when the average number of updates is not impacted much by a change in $K$, individual sequences can be impacted. Figure 7 illustrates this impact with a violin plot of the number of inputs processed per sequence for various $K$ and $L = 8$ on HAR. Decreasing $K$ below 3 impacts the maximum observed number of updates over the whole sequence. However, even changes above $K = 3$ have an impact. Even though the observed maximum number of updates does not change, the violin changes as some sequences have fewer inputs processed. This is due to the fact that the computational constraint is enforced over each window. So, for $K \leq 5$, some sequences are prevented from exceeding the computational budget in some windows of inputs.

It is perhaps surprising that SkipW can achieve both a higher accuracy and lower computational cost than SkipRNN and ThrRNN, as the former decides on the next $K$ inputs to process periodically whereas the baselines decide on the next input whenever it has processed one and so decide more often and with more information. They can thus adapt the sampling pattern faster. However, we have observed that these baselines tend to process one input regularly, even if there is no information in

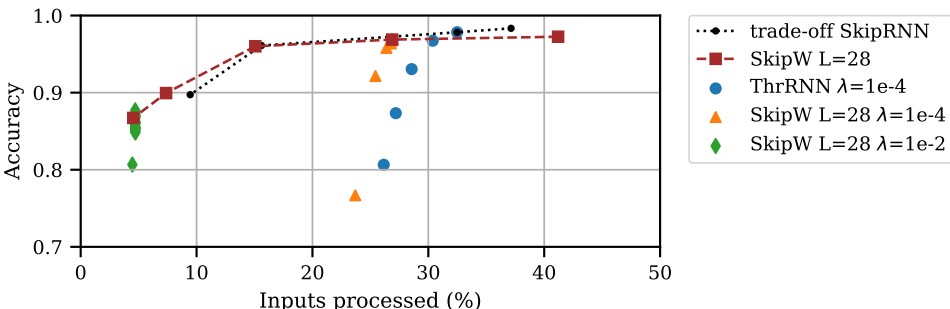

Figure 5: Impact of the parameter $K$ on accuracy and inputs processed for MNIST. Comparison between ThrRNN ($\lambda$=1e-4, $thr$=0.5 to 1.0) and SkipW ($\lambda$=1e-4 and $\lambda$=1e-2, $L$=28, $thr$=0.5, $K$=1 to $L$). The dotted line corresponds to the performance of individual SkipRNN models trained with different values of $\lambda$ (from $\lambda$=1e-4 to $\lambda$=1e-2). The dashed line represents the trade-off achieved by different SkipW models ($L$=28) (from $\lambda$=0 to $\lambda$=1e-2).

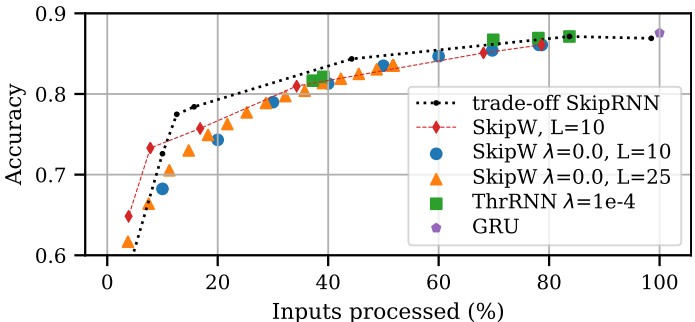

Figure 6: Impact of the parameter $K$ on accuracy and inputs processed for IMDB. Comparison between ThrRNN ($\lambda$=1e-4, $thr$=0.5 to 0.9) and SkipW ($\lambda$=0, $L$=10 or 25, $thr$=0.5, $K$=1 to $L$). The dotted (SkipRNN) and dashed (SkipW $L$=10) lines correspond to the performance of individual models trained with different values of $\lambda$ (from $\lambda$=0 to $\lambda$=1e-2)

the signal. This increases the computational cost without benefiting accuracy. We believe this is due to the increment mechanism (Equation 3). To skip $C$ inputs, $\Delta\tilde{u}_t$ (Equation 2) must be smaller than $0.5/C$. We conjecture that at some point, for large $C$, the model is unable to differentiate between these small values and so no more than $C$ skips are performed. So both methods are biased in the type of skip patterns they can learn, but differently. Please see Appendices D.3 and E for more details. The best method depends on the task: in our experiments SkipW seems to work better than SkipRNN on two data sets, similarly in one and slightly worse on IMDB.

SkipW is able to learn where relevant inputs are and tends not to use irrelevant inputs. This is best illustrated on AddingTask. Let $m_1$ be the index of the first marker, and $m_2$ the index of the second one. The inputs can be divided into four parts $p_i, i \in \{1, 2, 3, 4\}$:

$$p1 = (x_i \mid 0 \leq i \leq m_1) \qquad\qquad p2 = (x_i \mid m_1 < i < 25) \qquad (12)$$
$$p3 = (x_i \mid 25 \leq i \leq m_2) \qquad\qquad p4 = (x_i \mid m_2 < i < 50) \qquad (13)$$

Ideally the model should skip all inputs in $p_2$ and $p_4$ where there is no marker, while using all the inputs in $p_1$ and in $p_3$ where a marker can appear.

Figure 8 illustrates that SkipW skips inputs in different parts of the sequence for various $K$. For $K = L$ (training point), almost no inputs are skipped in $p_1$ and $p_3$, where the markers are, but some are skipped in $p_2$ and $p_4$, where there is no marker. Not all inputs are skipped in $p_2$ and $p_4$. That's not surprising: decisions to skip are taken at the start of a window, so even if a marker is present in the first input of the window, SkipW will not update its decision to process other inputs in the window. Skips are still mainly made in part $p_2$ and $p_4$ for these $K$ values but start increasing in $p_3$ as the model can no longer process all inputs. Finally, when $K < 5$, SkipW can no longer process the first 5 inputs and skips in $p_1$ start happening. Values of $K$ smaller than but close to $L$ force the

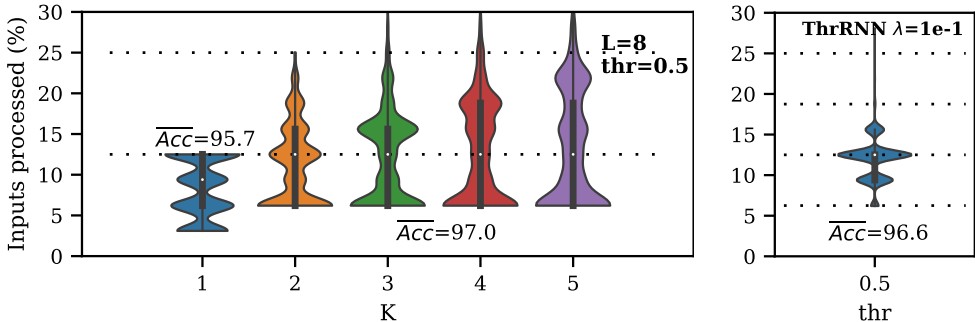

Figure 7: Left: Impact of the $K$ parameter on the computational cost upper limit (SkipW: $\lambda$=6e-2, $L$=8, $thr$=0.5). Right: ThrRNN with a similar average inputs processed is provided for comparison.

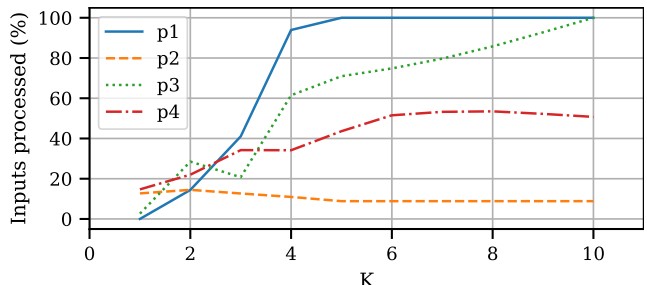

Figure 8: Proportion of inputs processed in Adding Task for different parts $p_i$ of the sequence defined by equations 12 and 13 while changing $K$ with $thr$=0.5, for a model trained with $\lambda$=2e-4 and $L$=10. The values at $K = L$ correspond to the training configuration.

model to skip some inputs and to 'fail' Adding Task. Note that both the binarization function and the inclusion of $t$ in the inputs of the update gate are necessary for this behavior. Please see the ablation study in Appendix D.4 for more information.

Modifying $thr$ impacts the accuracy / processing tradeoff as well. Increasing $thr$ gradually reduces both accuracy and average computational cost. This is similar to ThrRNN. Changing $thr$ does not affect the upper bound on computational cost but can provide more accuracy / processing tradeoff points. Some examples are provided in Appendix C.

## 4.2    MODIFYING THE TRADEOFF DURING INFERENCE

Modifying $(thr, K)$ is also possible during the analysis of a sequence. Results of a change between two different pairs of $(thr, K)$ values are shown in Figure 9. The average accuracy and computational cost of the analysis of the test sequences are reported as a function of the position of the change. The position at the bottom left (resp. top right) of the graph corresponds to the operating point of tradeoff$_2$ (resp. tradeoff$_1$) when it processes the whole sequence. The accuracy typically remains between the accuracy of the individual models and increases almost monotonically from tradeoff$_2$ accuracy to tradeoff$_1$ accuracy. This suggests that SkipW can switch between operating points during the analysis of a sequence. This behavior was observed on all data sets, for various training runs and $(thr, K)$ configurations. More details are provided in the supplementary material (D and F).

## 4.3    REAL TIME EXPERIMENTS

We evaluate the performance of SkipW on small hardware to showcase its performance in the setting it is designed for. We implement the full service, from images to activity recognition, on a Nvidia Jetson Nano platform. In the HAR-2D-POSE data set, OpenPose is used to compute poses on images, but the embedded GPU is not powerful enough to run this OpenPose. Instead, we choose to

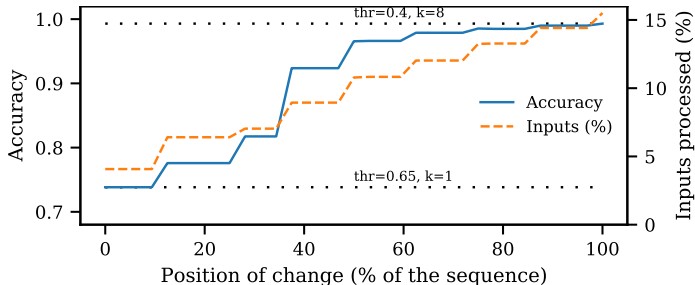

Figure 9: Modifying $(thr, K)$ during the sequence on HAR for SkipW ($\lambda$=1e-2, $L$=8): from tradeoff$_1$($thr$=0.4, $K$=8, acc/updates tradeoff = 99.3%/15.5%) to tradeoff$_2$($thr$=0.65, $K$=1, acc/updates tradeoff=73.8%/4%).

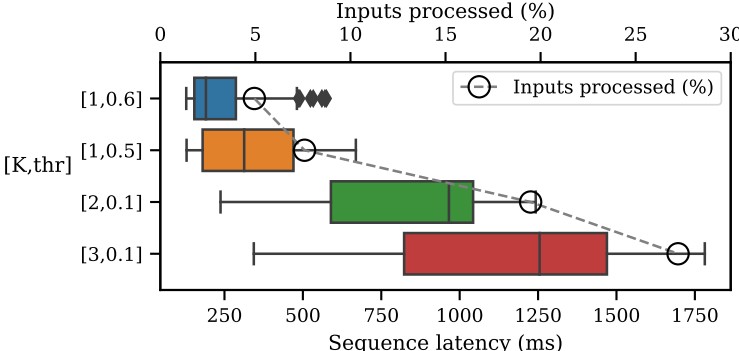

Figure 10: Boxplots of inference latency time for a HAR Service based on SkipW ($\lambda$=1e-2, $L$=8) coupled with a Posenet model running on Jetson Nano.

use PoseNet (MobileNetV1 architecture with a 0.75 multiplier) model which is much more efficient to run on such a small platform. In our implementation, PoseNet model running on the embedded GPU takes roughly 120 ms to compute a pose on a 640x480 pixel image, while the SkipW model takes around 30 ms to perform all state updates for a sequence.

Figure 10 presents the individual inference time of 1000 HAR image sequences processed by SkipW coupled with the PoseNet detection model on a Jetson Nano platform. As expected, when $K$ decreases, the run time decreases sharply. This gain is largely due to the lower number of pose detections that need to be computed. By drastically limiting the number of images used as input, SkipW saves a large amount of time, computation and energy. Results on Jetson TX2 and Raspberry Pi4 lead to similar conclusions (Appendix G).

## 5 CONCLUSION

We presented Skip-Window, a novel RNN cell architecture that dynamically controls the number of inputs processed during the analysis of a sequence. SkipW allows to adapt the computational cost during and between sequence analysis without reloading any weight. Unlike existing flexible RNNs, SkipW can enforce a strict upper bound on the number of updates performed over a time interval. Therefore, SkipW can operate without exceeding a given computational budget across a time interval. Furthermore, our results show that Skip-Window is able to exceed the accuracy of existing flexible RNNs for a lower computational cost and to actually reduce computation. This makes SkipW well suited for shared hardware with real time constraints.

ACKNOWLEDGMENTS

This work was supported by the European Commission under European Horizon 2020 Programme, grant number 951911 - AI4Media.

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

## A    EXPERIMENTAL SETTINGS DETAILS

We evaluate our approaches on two data sets:

- **Human Activity Recognition (HAR):** HAR is an action recognition dataset consisting of 2D pose estimations extracted from videos (Eiffert, 2018). The 2D pose estimation was performed on a subset of the Berkeley Multimodal Human Action Database (MHAD) dataset (Ofli et al., 2013). Six actions (Jumping in place, Jumping jacks, Punching (boxing), Waving two hands, Waving one hand, Clapping hands) are considered for a total of 1438 videos recorded at 22Hz. Each action must then be identified based on a sequence of 32 frames ($\approx$ 1.5 sec). Each video gives several sequences by applying an overlapping ratio. A pose is extracted from each frame and estimated by 18 body joints expressed in a 2D coordinate system. Following the parameter setting defined in (Eiffert, 2018), the dataset is split into 2 independent partitions : 22625 sequences for training and 5751 for validation.

  The model architecture consists of a two-stacked RNN of 60 GRU cells each followed by a fully connected layer with a RELU activation function. The model is trained with batches of 512 sequences using a decaying learning rate for 600 epochs. The best performance was obtained when the model was initialized with a model trained with a smaller $\lambda$.

  This architecture was selected by performing an optimization over the hyperparameters. The following parameters were included in the search:

  - Batch size: 4096 and 512
  - $\lambda \in \{1e-4, 1e-3, 1e-2\}$
  - Cell type: LSTM or GRU
  - Number of cells $\in \{30, 40, 50, 60\}$ per layer (identical number of cells in each layer)
  - Window size $L \in \{4, 8, 16\}$ (SkipW only)

  The search was performed both for ThrRNN (baseline) and SkipW (our approach). Several combinations of parameters resulted in models that are Pareto optimal with respect to accuracy and computational cost. To evaluate SkipW and compare it to ThrRNN, we selected one parameter configuration that led to models that were Pareto optimal (or close to Pareto optimal) for ThRNN and for several values of L for SkipW.

- **Adding Task:** This is a synthetic dataset consisting of sequences of (value, marker) tuples, where the expected output is $\sum_i marker \times value$. This dataset is commonly used to evaluate the performance of RNN models [(Hochreiter & Schmidhuber, 1997), (Neil et al., 2016), (Campos et al., 2018)]. The $value$ elements are uniformly sampled in the range (-0.5, 0.5) and the marker elements are equal to 1 or 0. We reproduced the experimental setup defined in (Neil et al., 2016) with a sequence of length 50. Two values are marked for addition with the first marker randomly placed in the first 10% of the sequence and the second one in the last half of the sequence. The variance of the task is equal to 1/6: the number of markers times the variance of the value (1/12). The task is considered solved when the average mean square error is lower than 1/100 the variance of the task.

  The model architecture consists of a single layer of 110 LSTM cells and a fully connected layer regressing the scalar output. We train the model with a batch size of 256 and a learning rate of $1e-4$. As with HAR dataset, the best performance was obtained when the model was initialized with a model trained with a smaller $\lambda$.

- Sequential **MNIST:** For this task the MNIST handwritten digits (Lecun et al., 1998) are flattened to vectors of length 784 and each pixel is presented sequentially to the RNN network. This becomes a challenging task for RNNs and is useful to test long range dependencies (Le et al., 2015). We follow the standard data split and set aside 5,000 training samples for validation purposes. The model architecture consists of a single layer of 128 GRU cells followed by a fully connected layer using the last hidden state to predict the digit. The model is trained for 600 epochs and we report results for the models achieving the best performance on the validation images.

- **IMDB:** This is a sentiment analysis task (Maas et al., 2011) using movie reviews extracted from the IMDB website. It consists in 25000 training and 25000 test reviews annotated as positive or negative. We set aside about 15% of training data for validation purposes. For this experiment we use the first 20000 most common words and a review length of

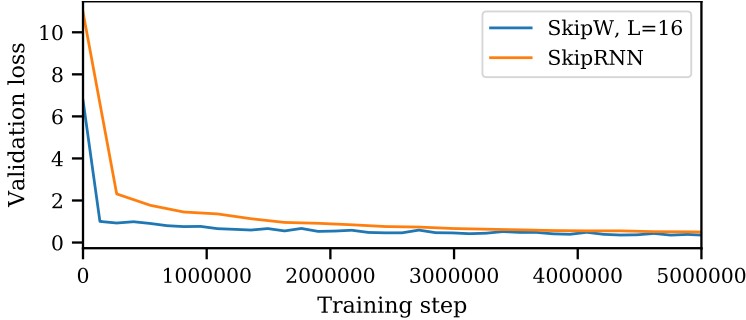

Figure 11: Validation loss during training for SkipW and SkipRNN on HAR. These two models use approximately the same number of inputs, respectively 11% and 13%.

200 by cropping or padding sequence accordingly. The model architecture consists in an embedding layer of size 300 initialized with GloVe embeddings (Pennington et al., 2014) or uniformly random vectors if unavailable. The embedded sequence are then passed to a 128 cell single layer GRU followed with a dropout (rate 0.2) before using a fully connected layer for the classification.

## B  TRAINING SPEED

There does not seem to be major difference between the training speed of SkipW and SkipRNN, as is illustrated in Figure 11 for two models using approximately the same number of updates. Convergence speed is however affected by the choice of $\lambda$. Higher $\lambda$ constrain the models more and therefore lead to slower training.

During training, to process inputs in batch, the model uses the implementation described in Section 3, based on the multiplication by $u_t$ and $(1 - u_t)$ rather than actually skipping inputs. This allows the use of matrix operations on mini-batches of inputs, which makes training faster than processing sequences one by one and skipping inputs.

## C  MODIFYING THR

Modifying $thr$ impacts the accuracy / processing tradeoff as well. However, changing $thr$ does not affect the upper bound on computational cost. Increasing $thr$ gradually reduces both accuracy and average computational cost. This is similar to ThrRNN. Figure 12 illustrates the impact of changing $thr$ for various $L$ on HAR. While this is less interesting in the context of a strict limit on computation, it increases the variety of the possible tradeoff points. That being said, the thresholding mechanism allows the model to skip uninteresting inputs and is therefore necessary. For example, if this mechanism is removed on adding task, SkipW processes inputs in parts of the sequence where there is no marker. On the other tasks, removing it similarly increases computational cost without improving accuracy. This is better highlighted in Appendices D.4 and H.2.

## D  DETAILED RESULTS FOR ADDING TASK

The Adding task dataset penalizes the model when skipping markers. Therefore, a successful model will learn to process as many inputs as possible where the makers can be placed and to ignore the parts of the sequence where the markers are never found.

### D.1  ACCURACY AND COMPUTATIONAL TRADEOFFS

As we see in Figure 4, both SkipW and ThrRNN offer operating points that are better than the ones obtained with individual SkipRNN models (dotted line in the figure). SkipW offers results with a

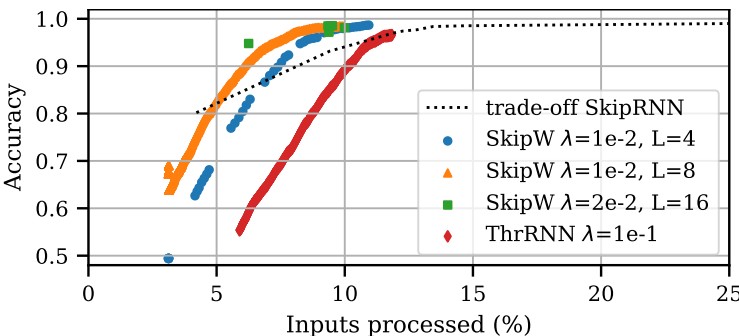

Figure 12: Impact of the thr parameter on accuracy/updates tradeoff for HAR. Comparison between ThrRNN ( $thr$=0.5 to 1.0), SkipW ($thr$=0.5 to 1.0, $K = L$) and SkipRNN ($\lambda$=1e-3 to 3e-1).

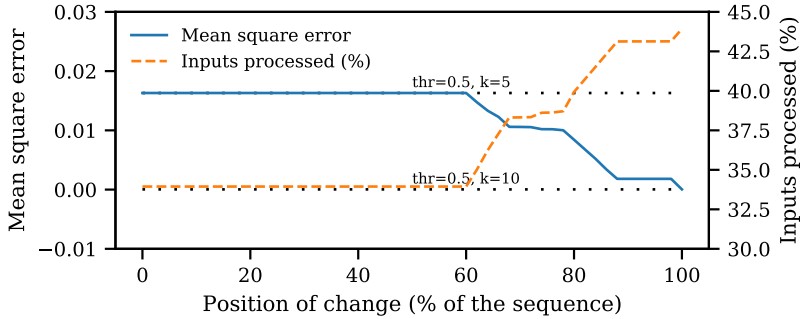

Figure 13: Modifying $K$ during the analysis of a sequence on Adding Task for SkipW ($\lambda$=2e-4, $L$=10): from tradeoff$_1$($thr$=0.5, $K$=10, MSE/updates = 3.56e-5/44.4%) to tradeoff$_2$($thr$=0.5, $K$=5, MSE/updates=1.66e-2/34.2%).

lower number of updates and lower error than ThrRNN while adapting $thr$ especially for $K$ between 5 and 10. Both models also achieve a lower error than a model randomly sampling inputs.

## D.2 MODIFYING TRADEOFF DURING INFERENCE

We test SkipW adaptability during the analysis of a sequence; switching between a model with a low MSE, high update rate and another model with a higher MSE and lower update rate.

As illustrated by Figure 13, the evolution of the MSE is not as smooth as in the HAR experimentation due to the nature of the task - skipping a marker greatly penalizes the MSE. We can note that changes in MSE and number of inputs processed happen when the switch is performed in the last 40% of the sequence. This is explained by the fact that the distribution of the inputs processed is identical for $K = 5$ and $K = 10$ at the beginning of the sequence. Both models a) process the first 5 inputs, to find the first marker, b) skip other inputs up to 50% of the sequence, as they never contain a marker and c) process the last 5 inputs of the window containing the middle of the sequence, as the second marker may be there. This is illustrated in Figure 16 for $K = 10$ and Figure 18 for $K = 5$.

As the loss in MSE stays reasonable proportionally to the number of updates skipped, SkipW can be used to reduce the number of updates during a sequence to fulfill computational restrictions.

## D.3 SKIP PATTERNS AND COMPARISON TO SKIPRNN AND THRRNN

Figure 14 illustrates the ability of SkipW to strictly limit the number of inputs processed. The dotted lines corresponds to the $K/L$ theoretical limit imposed. It is worth noting that the model uses a maximum of 60% of inputs for $K = L$. This can be explained by the nature of the task where 40% of the sequence does not contain information (no marker) and is also illustrated by the distribution of inputs processed in Figure 16 (SkipW, top row).

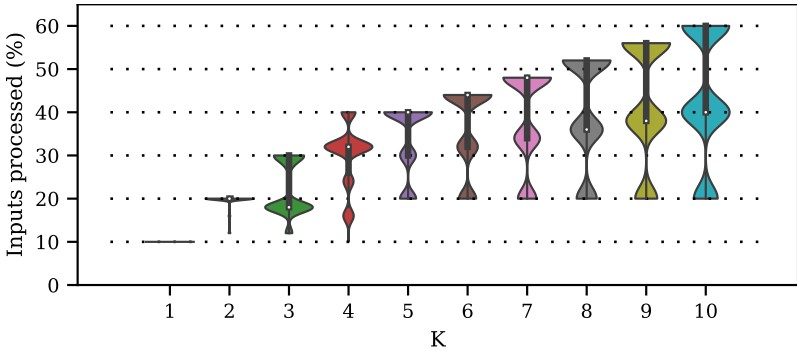

Figure 14: Impact of the $K$ parameter on the computational cost upper limit (SkipW: $\lambda$=2e-4, $L$=10, $thr$=0.5).

During training, there is no restriction on the value of $K$ and no restriction on the patterns SkipW can explore and therefore it can discover any skipping scheme. Figure 16 illustrates that an unconstrained ($K = L$) SkipW has correctly learned that it must sample the first 10% of the sequence and then stop (until the second marker can appear). It has also learned that it must resume sampling when the second marker can appear. SkipW has also learned that it can stop sampling the end of the sequence when the second marker is detected: the proportion of skips increases in each window after the third one.

When a computational constraint is enforced and $K$ is set to a lower value, some skip patterns are forbidden and skipW falls back to other patterns. These restrictions inevitably cause SkipW to miss some markers and therefore to fail the task. This is illustrated in Figure 15. If failing the task is not an option, one should not use $K < L$. Note that SkipW has learned to fall back to interesting sampling patterns: the first marker starts to be skipped only when $K < 5$.

Figure 16 also compares the SkipW models with ThrRNN/SkipRNN. This figure highlights the respective advantages of each method when flexibility is put aside ($K = L$ and $thr$=0.5). Because the skip mechanisms are different for SkipW and SkipRNN, these methods produce different sampling patterns. SkipW determines the inputs to process at the beginning of each window whereas ThrRNN performs a decision whenever an input is processed. ThrRNN is therefore able to stop sampling inputs when it has detected a marker. For example, the probability of processing an input gradually decreases after position 25 for ThrRNN making it more efficient for the second part of the sequence as shown in Table 1. The same behavior can be observed for SkipW, but only from one window to another. It is also worth noting that the probability that SkipW processes inputs in a window is very close to the probability that ThrRNN processes the first input in the window. On the positive side for SkipW, notice that ThrRNN is unable to stop processing inputs for a long period, as shown by a non-zero probability of using inputs between position 5 and 25. In addition, ThrRNN sometimes skips the first potential location of the second marker. We believe this explains why SkipW can achieve both a higher accuracy and a lower computational cost than SkipRNN and ThrRNN. Because ThrRNN is dragged down by its inability to completely stop sampling where there is no information, when trying to further reduce the number of inputs processed by increasing $\lambda$, skipRNN stops analyzing interesting inputs.

This can also been seen by looking at sampling patterns, summarized in Figure 18 for some values of $K$, or on MNIST in Appendix E.

## D.4   IMPACT OF ARCHITECTURE ELEMENTS ON THE DISTRIBUTION OF INPUTS PROCESSED IN A SEQUENCE

### D.4.1   ABLATION OF INDEX IN SEQUENCE (T)

As discussed in Section 3, including the index ($t$) in the input of the update gate (Equation 6) is necessary to allow the model to skip all inputs in a window. We show here the impact of removing it by modifying equation 6: $\tilde{u}_{W,t+1} = \gamma \cdot \sigma(W_w(s_{t-1}) + b_w) + (1 - \gamma) \cdot \tilde{u}_{W,t}$.

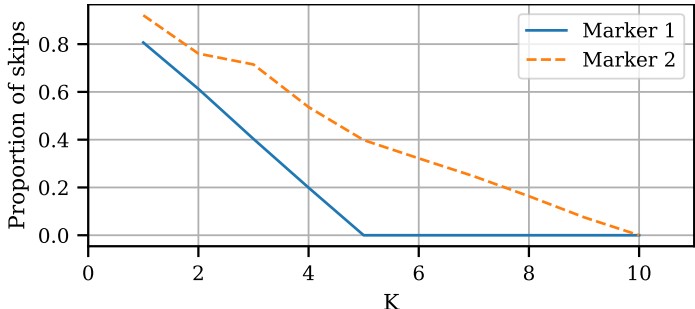

Figure 15: Impact of the $K$ parameter on the proportion of markers skipped. (SkipW: $\lambda$=2e-4, $L$=10, $thr$=0.5).

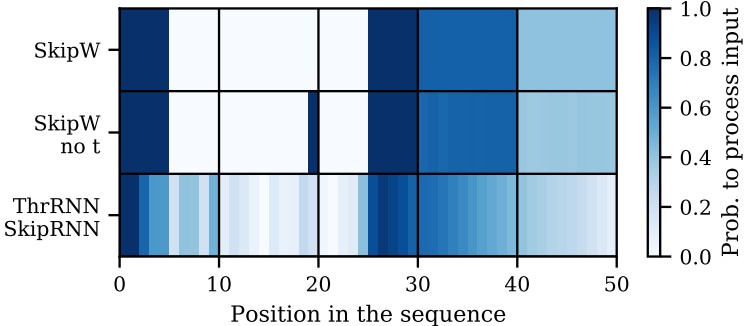

Figure 16: Distribution of the inputs processed for Adding Task using different models with $thr$ = 0.5. From top to bottom: SkipW $K = L = 10$ and $\lambda$=2e-4, SkipW with no index in sequence $K = L = 10$ and $\lambda$=3e-4, ThrRNN $\lambda$=3e-4.

Figure 16 shows that, without $t$, the model is forced to process an additional input in the second window even though this part of the sequence contains no marker. Processing this input enables the model to change its hidden state and therefore to obtain a different activation vector $\tilde{u}_{W,t+1}$ in the third window and to continue processing the second part of the sequence where the second marker is found. Without this update, the model would stop processing the inputs at the end of the first window since the state and therefore the activation vector would not change. Including $t$ in the input of the activation gate ensures that a different activation vector may be computed even without processing an input in a window.

### D.4.2 ABLATION OF THRESHOLD (THR = 0)

A second experiment was done to show the importance of the threshold parameter. When the threshold is set to 0 the inputs are selected sorely on the basis of the parameter $K$. There-fore exactly $K$ inputs are selected in each window using equation 9 and equation 10 becomes $u_t = f_{binarize}(\tilde{u}^K_{W,t}[i]) = 1$ if $\tilde{u}^K_{W,t}[i] > 0$, 0 otherwise.

Figure 17 shows the distribution of the inputs in each window for different values of $K$ when $thr = 0$ and Figure 18 when $thr = 0.5$. The importance of $thr$ is clearly highlighted when comparing the two figures. For example, for $K = 5$ and $thr = 0$ (Figure 17), there is no difference in the first window and the model finds the first marker. However, as the model is forced to use 5 inputs per window, it processes inputs in the second window and this impacts the choice made in the third

Table 1: Inputs processed in different part of sequence for Adding Task

| Architecture | first half | second half |
|---|---|---|
| SkipW ($L = 10, \lambda = 2e - 4$) | 20% | 69% |
| ThrRNN ($\lambda = 3e - 4$) | 31% | 54% |

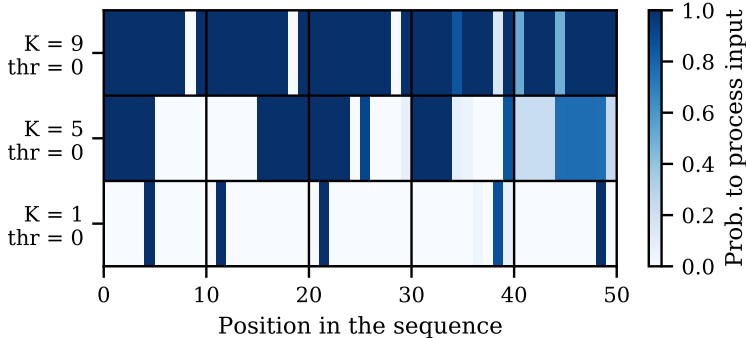

Figure 17: Distribution of the inputs processed for Adding Task using a SkipW model trained with $thr$=0.5, $L$=10 and $\lambda$=2e-4; inference performed with $thr$=0 and different values of $K$.

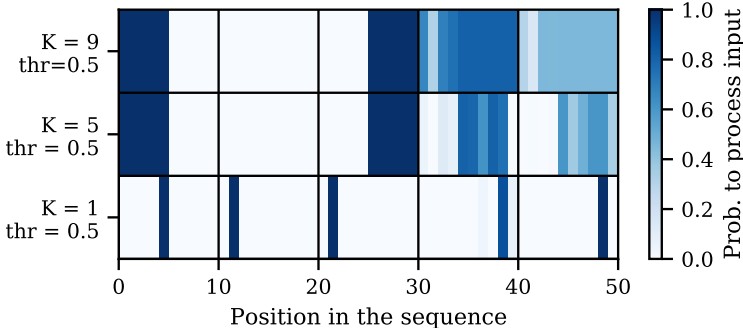

Figure 18: Distribution of the inputs processed for Adding Task using a SkipW model trained with $L$=10 and $\lambda$=2e-4 with $thr$=0.5 and different values of $K$.

window as the model processes inputs at the beginning of this window instead of the last 5 which could contain the second marker. This behavior is to be compared with Figure 18 ($thr = 0.5$) where the inputs processed are effectively located in the second half of the third window. It should be noted that for $K = 1$, $thr$ has no discernible effect. The model selects only one input in the first window and this affects the performance (the model will miss the first marker 80% of the time) but also which inputs are selected for processing in subsequent windows.

## E  DETAILED RESULTS FOR SEQUENTIAL MNIST

To better assess the performance of SkipW on long sequences, we apply it to sequential MNIST. In this data set, pixels of 28×28 images are processed sequentially. We selected $L = 28$, which means each window corresponds to a line of the image. Some results are available in Table 2 and Figure 19. The first part of the table contains methods processing pixels sequentially. The second part reports recent results from feature selection methods. SkipRNN by Campos et al. (2018) was reported with an accuracy of 97.6% when processing half the pixels in the original paper. We generated additional tradeoff results with SkipRNN by gradually increasing $\lambda$.

We report results for skipW models using different $\lambda$ from 1e-4 to 1e-1. For $K = L$, skipW achieves similar tradeoffs to skipRNN models. The number of inputs and accuracy can be modified with $K$. However, on this dataset, SkipW accuracy decreases faster with $K$ and the resulting tradeoffs do not stay above the ones achieved by other SkipRNN models. For example, SkipRNN ($\lambda$=1e-3) achieves both a better accuracy and computational cost than SkipW ($L$=28, $K$=9, $\lambda$=1e-4). If reloading the model is an option, selecting another model may be better than modifying $K$. However, doing so with SkipRNN would not enforce any computational constraint while SkipW can enforce rather tight constraints. For example, using SkipW ($L$=28, $\lambda$=1e-4) $K$ can be decreased to 11 before significantly affecting accuracy. Similarly, SkipW ($L$=28, $\lambda$=1e-2) processes on average only 4.5% inputs to achieve 86.7% accuracy and can be limited to 3 pixels per line without impacting

Table 2: Comparison of various subsampling methods on MNIST. The first part of the table contains methods processing pixels sequentially. The second part contains some results from recent feature selection methods. These are not sequential.

| Model | Source | Accuracy (%) | Inputs processed (%) |
|---|---|---|---|
| RNN ($p_{skip} = 0.5$) | Campos et al. (2018) | 91.2 | 50 |
| SkipRNN ($\lambda =$1e-4) | Campos et al. (2018) | 97.6 | 50 |
| SkipRNN ($\lambda =$1e-4) | | 97.8 | 32.5 |
| SkipRNN ($\lambda =$1e-3) | | 96.1 | 15.6 |
| SkipRNN ($\lambda =$1e-2) | | 89.7 | 9.5 |
| SkipRNN ($\lambda =$1e-1) | | 10.0 | 0.7 |
| SkipW ($L$=28, $K$=28, $\lambda =$1e-4) | | 96.9 | 26.9 |
| SkipW ($L$=28, $K$=11, $\lambda =$1e-4) | | 96.4 | 26.8 |
| SkipW ($L$=28, $K$=10, $\lambda =$1e-4) | | 95.8 | 26.4 |
| SkipW ($L$=28, $K$=9, $\lambda =$1e-4) | | 92.2 | 25.4 |
| SkipW ($L$=28, $K$=8, $\lambda =$1e-4) | | 76.7 | 23.7 |
| SkipW ($L$=28, $K$=7, $\lambda =$1e-4) | | 57.5 | 21.3 |
| SkipW ($L$=28, $K$=28, $\lambda =$1e-3) | | 96.0 | 15.1 |
| SkipW ($L$=28, $K$=28, $\lambda =$1e-2) | | 86.7 | 4.5 |
| SkipW ($L$=28, $K$=3, $\lambda =$1e-2) | | 86.7 | 4.5 |
| SkipW ($L$=28, $K$=2, $\lambda =$1e-2) | | 83.0 | 4.3 |
| SkipW ($L$=28, $K$=28, $\lambda =$1e-1) | | 10.0 | 0.1 |
| Uniform sampling | Huijben et al. (2019) | 71.4 | 3.2 |
| DPS-topK | Huijben et al. (2019) | 93.4 | 3.2 |
| LOUPE (Bahadir et al., 2019) | Huijben et al. (2019) | 80.1 | 3.2 |
| Concrete Autoencoders | Balın et al. (2019) | 90.6 | 6.4 |

accuracy much. That being said, its operating range is a bit smaller than the first model. To put the performance of this model in perspective, the second part of Table 2 contains results achieved by recently published feature selection methods on the MNIST data set. These methods consider all pixels at the same time for both feature selection and classification. They are therefore not directly comparable but they offer an interesting point of comparison. The accuracy (86.7%) and inputs processed (4.5%) achieved by the SkipW model ($L$=28, $\lambda$=1e-2) are not as good as the best of these methods (DPS-TOPK). They are however much closer to these results than to these based on uniformly sampled pixels and arguably better or similar to the 80.1% accuracy for 3.2% inputs processed of LOUPE.

This dataset also further shows that the skip patterns learned by SkipW and SkipRNN can be biased. Figures 23 to 26 illustrate skip patterns realized by SkipW on some digits. They can be compared to the results achieved with SkipRNN, displayed in Figures 20 to 22. A few interesting observations can be made about SkipW. First, most pixels are sampled in the middle of the image, where the digits are located. No pixels are looked at in the top, left or right of the pictures but several are considered at the bottom. SkipRNN on the other hand samples a few pixels in the top and bottom of the figures and a lot on the sides.

Then, SkipW patterns are characterized by vertical lines: the same positions tend to be sampled in each window. These positions are not all static and vary depending on the particular sequence. Furthermore, when $K$ is small, the pixels sampled tend to be spread out in the image. Finally, Figures 25 and 26 further support that SkipW has learned interesting patterns in the data. For example, for digits 1 and 0, sampling almost exclusively takes place in one column, with some additional pixels next to the top of that line. We presume these are useful to check for a vertical or horizontal line there. This would indicate the digit is not a 1 or 0. For other digits, more pixels are sampled at the left of the main line. This can be compared with the SkipRNN sampling patterns. They also display a periodic behavior, but that periodicity is at the pixel rather than at the window level. While this bias seem shared by both methods, we conjecture that SkipW suffers less because it takes place at a larger scale. This may also explain why this behavior was not present for SkipW in Adding Task.

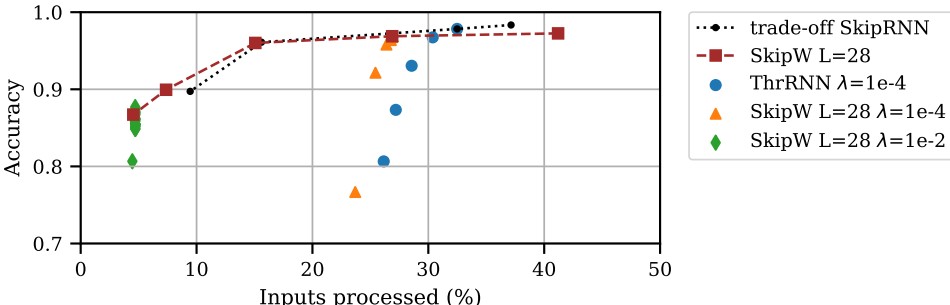

Figure 19: Comparison of SkipRNN and SkipW and impact of the $K$ parameter on accuracy/updates tradeoff for MNIST. For both SkipW ($L$=28) and SkipRNN several values of $\lambda$ were tested: from $\lambda$=1e-4 to $\lambda$=1e-2. This figure is the same as Figure 5 and illustrate some results listed in Table 2.



Figure 20: Examples of skip patterns by SkipRNN on MNIST with $\lambda$=1e-4. Blue pixels are skipped. This figure is from (Campos et al., 2018).

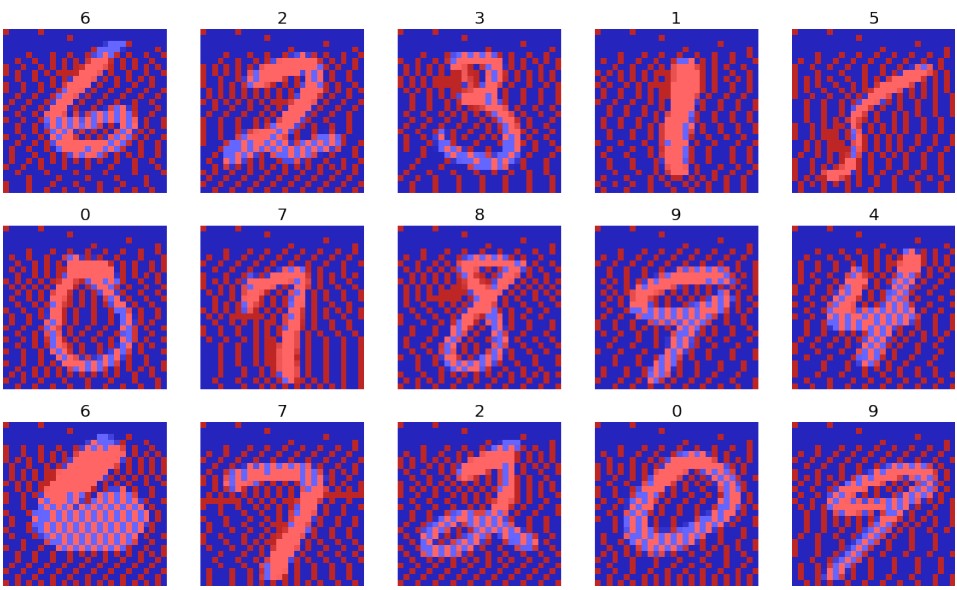

Figure 21: Examples of skip patterns by SkipRNN on MNIST with $\lambda$=1e-4. Blue pixels are skipped.

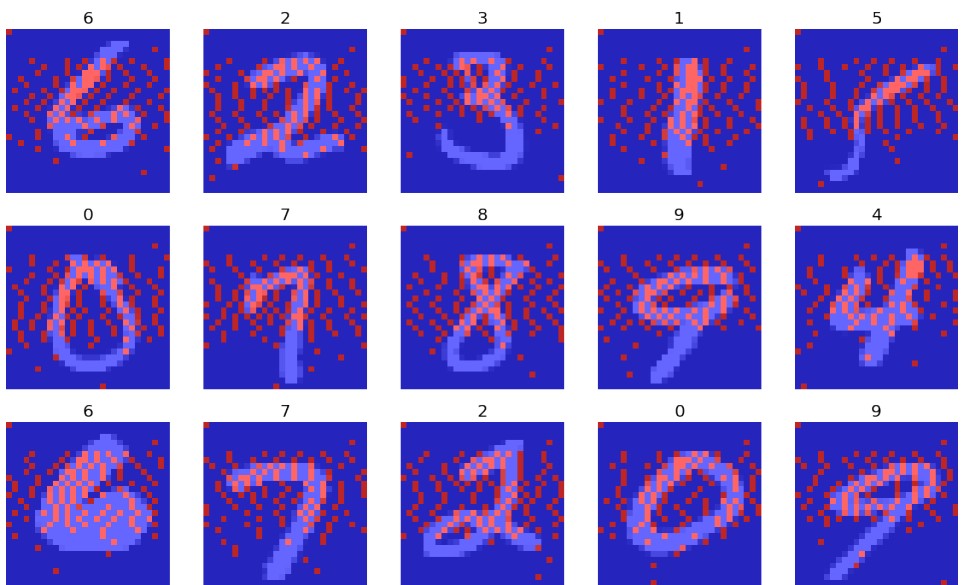

Figure 22: Examples of skip patterns by SkipRNN on MNIST with $\lambda$=1e-3. Blue pixels are skipped.

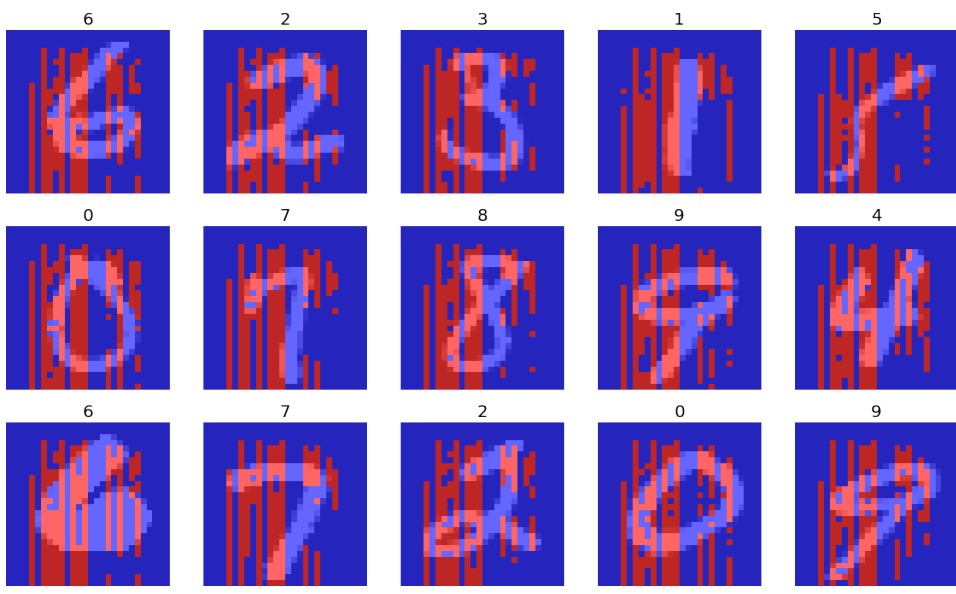

Figure 23: Examples of skip patterns by SkipW ($\lambda$=1e-4) on MNIST with $L$=28 and $K$=28. This particular model achieves an accuracy of 96.9% and uses on average 26.9% of inputs. Blue pixels are skipped. All images are correctly classified.

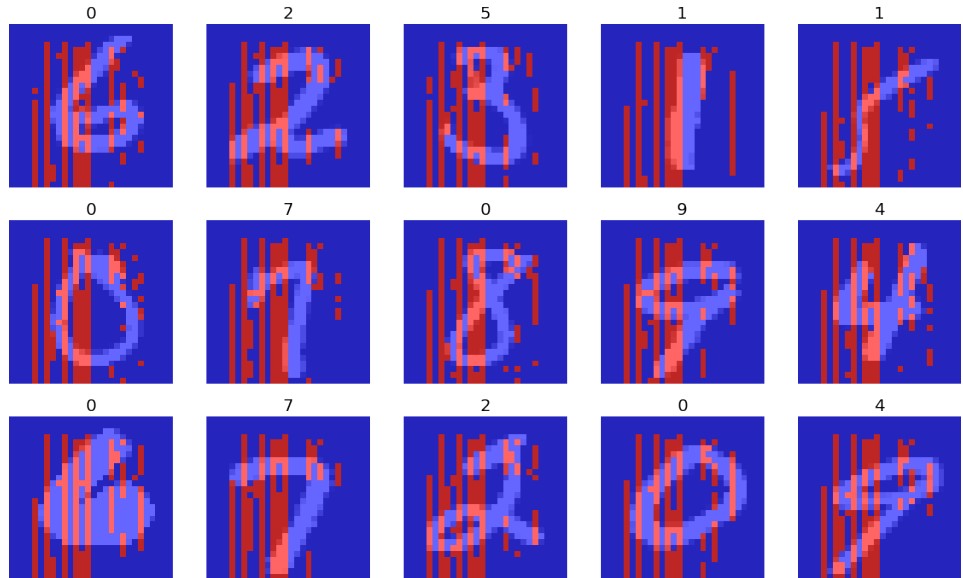

Figure 24: Examples of skip patterns by SkipW ($\lambda$=1e-4) on MNIST with $L$=28 and $K$=7. This particular model achieves an accuracy of 57.4% and uses on average 21% of inputs. Blue pixels are skipped. The predicted label is on top of each image.

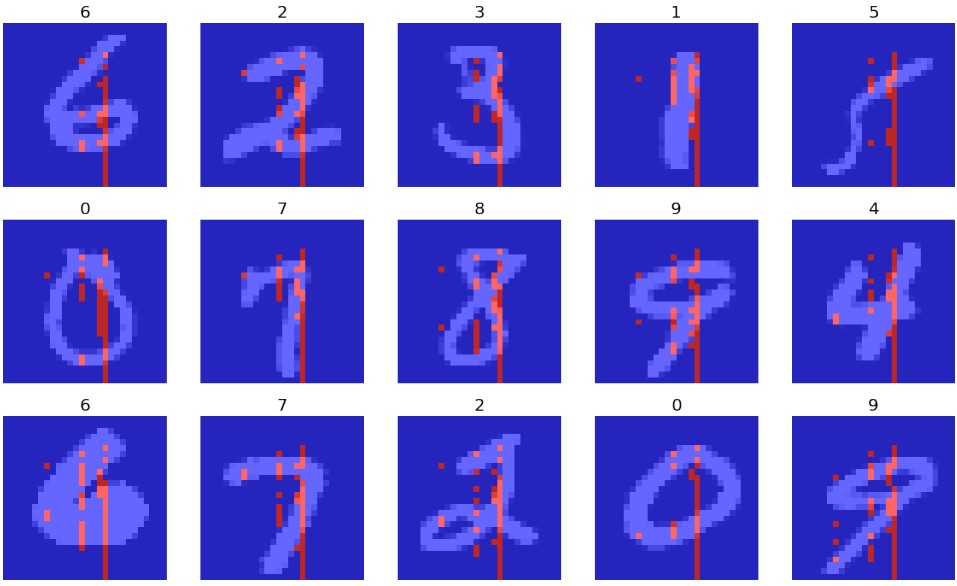

Figure 25: Examples of skip patterns by SkipW ($\lambda$=1e-2) on MNIST with $L$=28 and $K$=28. This particular model achieves an accuracy of 86.7% and uses on average 4.5% of inputs. Blue pixels are skipped. The predicted label is on top of each image.

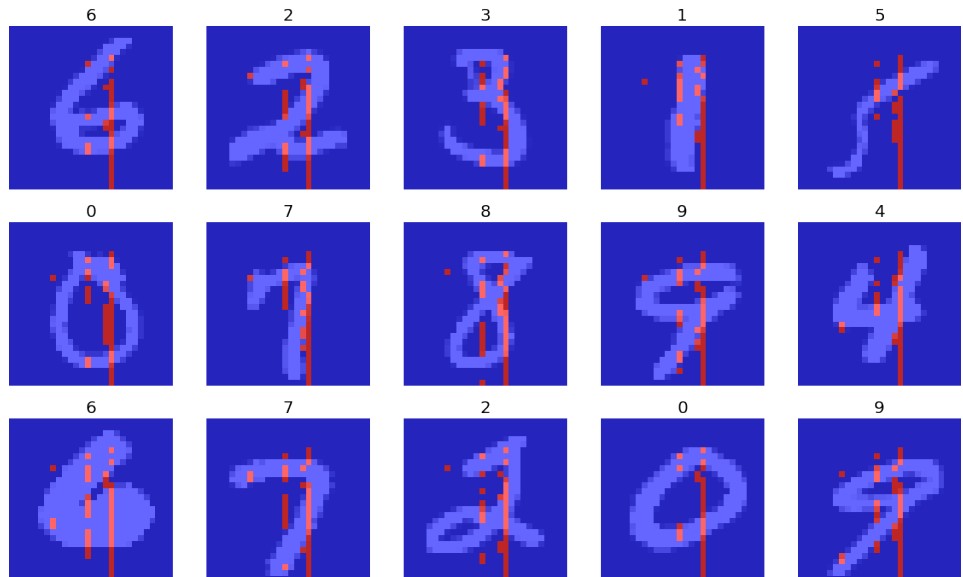

Figure 26: Examples of skip patterns by SkipW ($\lambda$=1e-2) on MNIST with $L$=28 and $K$=2. This particular model achieves an accuracy of 83% and uses on average 4.3% of inputs. Blue pixels are skipped. The predicted label is on top of each image.

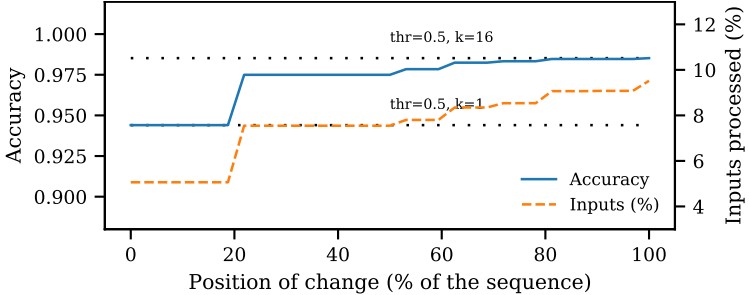

Figure 27: Modifying $(thr, K)$ during the sequence on HAR for SkipW ($\lambda$=2e-2, $L$=16): from tradeoff$_1$: ($thr$=0.5, $K$=16, acc/updates = 98.5%/9.5%) to tradeoff$_2$: ($thr$=0.5, $K$=1, acc/updates = 94.4%/5.1%).

## F  ADDITIONAL RESULTS FOR HAR

### F.1  MODIFYING TRADEOFF DURING INFERENCE

Figure 27 and 28 show the adaptability of SkipW during the analysis of a sequence when switching between different tradeoffs. We present here results for a SkipW wih $L = 16$ and $\lambda$=2e-2 with different tradeoffs. As with previous experiments (Figure 9 and Figure 13), the performance is bounded by the individual performance of the two configurations evaluated.

## G  REAL TIME EXPERIMENTS

### G.1  LATENCY EXPERIMENTS

In order to explore the impact of the HW platform on the latency of HAR Services based on SkipW/PoseNet, we measure the latency of the HAR Service on 2 other small platforms: a Jetson TX2 (similar to the Jetson Nano platform (Figure 10), but more powerful) and a Raspberry Pi 4 (better CPU, but lacking a compatible GPU for AI model acceleration). The hardware specification of these different devices is provided in Table 3. Figure 29 compares the performances of the HAR

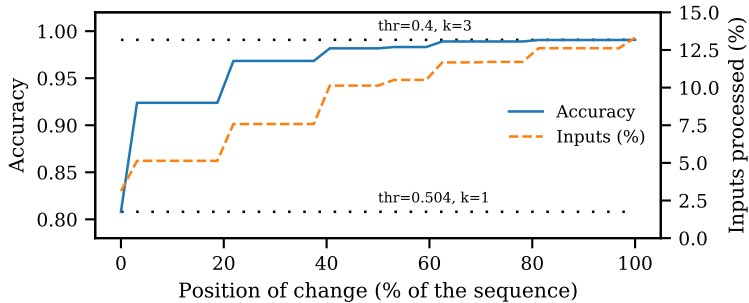

Figure 28: Modifying $(thr, K)$ during the sequence on HAR for SkipW ($\lambda$=2e-2, $L$=16): from tradeoff$_1$: ($thr$=0.4, $K$=3, acc/updates = 99.1%/13.3%) to tradeoff$_2$: ($thr$=0.504, $K$=1, acc/updates = 80.8%/3.1%).

Table 3: Specifications of hardware devices

|  | Jetson TX2 | Jetson Nano | Raspberry Pi 4 |
|---|---|---|---|
| AI performance | 1.33 TFLOPs | 0.47 TFLOPs | 0.0135 TFLOPs |
| GPU[1] | 256-core NVIDIA Pascal™ | 128-core NVIDIA Maxwell™ | - |
| CPU | 4-core ARM® A57 (@1.2GHz) & 2-core NVIDIA Denver (@854MHz) | 4-core ARM® A57 (@1.2GHz) | 4-core ARM® A72 (@1.5GHz) |
| Memory | 4 GB 128-bit LPDDR4 51.2 GB/s | 4 GB 64-bit LPDDR4 25.6 GB/s | 4 GB LPDDR4-3200 |

[1] Only AI capable GPUs are reported

Services on the 3 selected platforms. It is worth noting that the impact of the different tradeoffs ($[K, thr]$) on the latency are very similar.

## G.2 ENERGY CONSUMPTION

When SkipW coupled with a CNN model is set appropriately, it can enable a huge reduction in the amount of computation. This can have an immediate impact on the energy consumed to deliver a Service, an important feature for battery powered systems. As an illustration, we measure the power consumed by the HAR analysis when running on the Jetson boards. This analysis is designed to run at 22Hz. The energy consumption is measured by pulling periodically (every 100 ms) the instantaneous power consumption indicator available at the board level. We report here an average of these instantaneous values over a 8 minutes period. Figure 30 provides those energy consumption for the Jetson Nano and the Jetson TX2. Those two platforms display the same behavior: when $K$ decreases, the consumption decreases significantly. The gain is especially noticeable for the Jetson Nano platform, where it can reach up to 50% of energy gain.

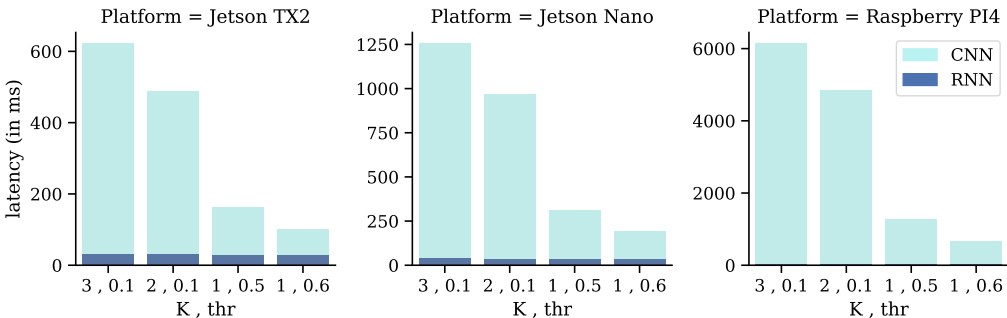

Figure 29: Sequence latency for different devices when processing HAR Service using SkipW ($\lambda$=1e-2, $L$=8) coupled with PoseNet (MobileNetV1 0.75). Individual contribution of models (CNN: PoseNet, RNN: SkipW) are also reported.

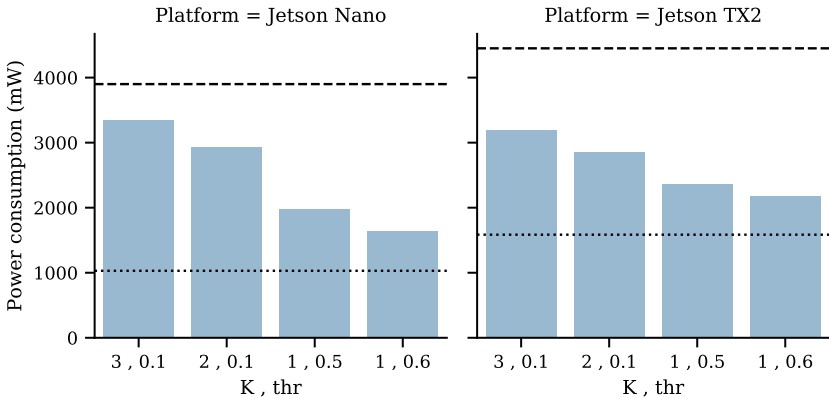

Figure 30: Average energy consumption for the analysis of a HAR sequence using SkipW ($\lambda$=1e-2, $L$=8) and PoseNet (MobileNet 0.75) on a Jetson Nano and a Jetson TX2. The dotted line corresponds to the energy level for no activity and the dashed one to the maximum instantaneous level measured when models are running.

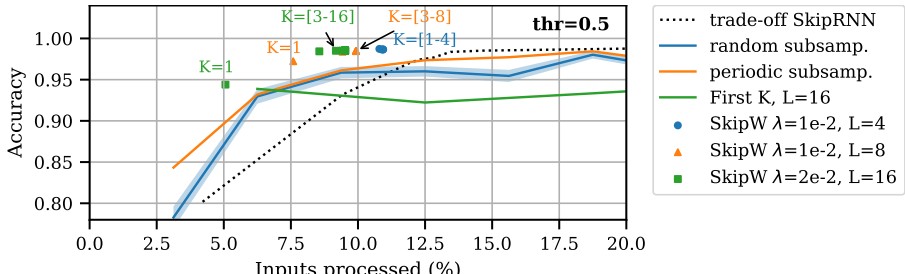

Figure 31: Impact of the $K$ parameter on accuracy/updates tradeoff for HAR. Comparison between ThrRNN ($thr$=0.5 to 1.0), SkipW($L$=4,8,16; $thr$=0.5, $K$=1 to $L$) and SkipRNN ($\lambda$=1e-3 to 3e-1). For random subsampling, the shaded area corresponds to 3 times the standard deviation on each side of the mean value (50 evaluations).

## H   WOULD OTHER SAMPLING MECHANISMS WORK JUST AS WELL?

The main purpose of SkipW is to upper bound the computational complexity of the model dynamically during inference. This has been illustrated in the experiments. Here, we study whether naive approaches would work just as well or whether some parts of the model can be modified. We conducted three sets of experiments on HAR. In the first one, SkipW is compared to naive models with a set complexity, such as random subsampling. In the second and third ones, we modify the skip mechanism of SkipW to generate other flexible models. To do so, we replace the sampling mechanism by random sampling variants and we respectively keep or discard the selection by the binarization function.

We selected HAR because SkipW works best on this data set. Furthermore, as information is located in only some parts of the sequence for Adding Task and MNIST, it seems obvious that naive sampling strategies would not work well on these data sets.

### H.1   NAIVE RANDOM SAMPLING

Figure 31 compares the models presented in the main experimental section to additional baselines. In addition to ramdomly sampling inputs, we also consider sampling inputs at set locations, such as the $K$ first inputs of each window or periodic sampling. When the number of inputs subsampled is small, no variant considered in this section achieves an accuracy similar to SkipW for a set number of inputs processed. However, they are more accurate than SkipRNN.

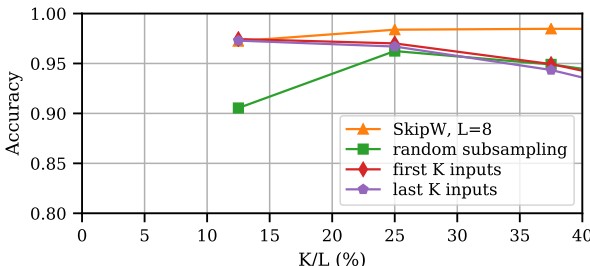

Figure 32: Replacing the sample mechanisms by other variants (without the binarization function) decreases accuracy on HAR with $K$=8.

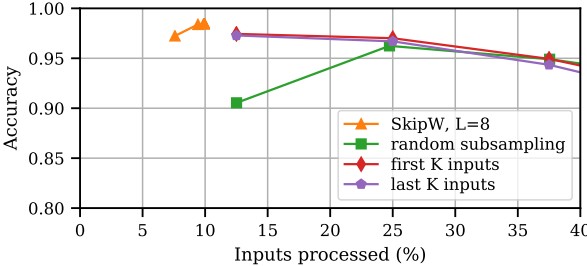

Figure 33: Replacing the sample mechanisms by other variants (without the binarization function) negatively impact the number of inputs processed on HAR with $K$=8.

## H.2  OTHER SAMPLING MECHANISMS, WITHOUT THE BINARIZATION FUNCTION

In this section we compare the SkipW architecture described in Section 3 to variants where the sampling mechanisms is modified to sample $K$ inputs within each window at random or by selecting the first or the last elements of the window. The results are displayed in Figures 32 and 34 for two different SkipW models ($L$=16 and $\lambda$=2e-2; $L$=8 and $\lambda$=1e-2). No variant matches the accuracy of the original SkipW over the whole range of $K$, although taking the first or last element achieves a similar performance for $K$=1 and so does random sampling for $K$ between 4 and 6 when $L$=16. That being said, SkipW can process fewer than $K$ inputs in every window thanks to the importance vector and the binarization function. Not using these negatively impacts the number of inputs processed, as shown in Figures 33 and 35. This leads to a big difference between the original SkipW architecture and the other variants studied here. Even in the most favorable case where $L = 16$ and $K = 1$, skipW processes 5.06% of inputs whereas the other variants process 6.25%. We expect this computational advantage to be even larger on tasks such as Adding Task and sequential MNIST, where a lot of windows contain no useful information.

The fact that no method matches the accuracy of SkipW for all $K$, that for some $K$ no method matches the accuracy of SkipW and that SkipW also process fewer samples than these variants show the interest of using the weight vector to select future inputs.

## H.3  OTHER SAMPLING MECHANISMS, WITH THE BINARIZATION FUNCTION

Finally, in this section we compare the SkipW architecture to an additional variant where the $select_K$ mechanism is modified to select the first $K$ inputs of the window whose associated value is $> thr$. In other words, the main difference with the previous section is that only TopK is replaced but the binarization function is kept. Figures 36 and 37 illustrate the results for $L = 8$ and 16 respectively. The previous section has shown that using the importance vector is necessary. This section suggests that the method used to select inputs from these that beat the threshold can change without impacting the accuracy when $thr = 0.5$. However, one should note that selecting inputs within each window without using the binarization function (previous section) is equivalent to using $thr = 0$. As the alternative $select_K$ variants were less effective than TopK for this configuration, we advice against not using TopK when modifying $thr$.

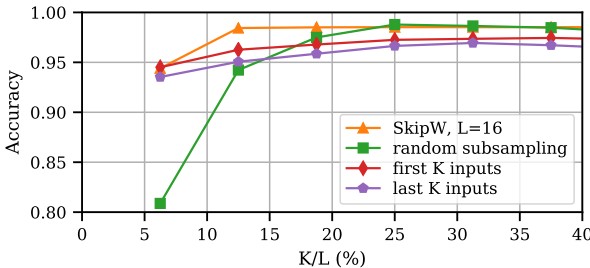

Figure 34: Replacing the sample mechanisms by other variants (without the binarization function) decreases accuracy on HAR with $K$=16.

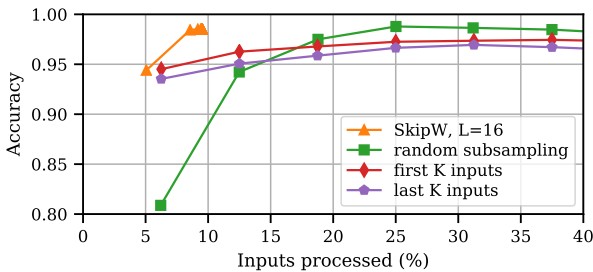

Figure 35: Replacing the sample mechanisms by other variants (without the binarization function) negatively impact the number of inputs processed on HAR with $K$=16.

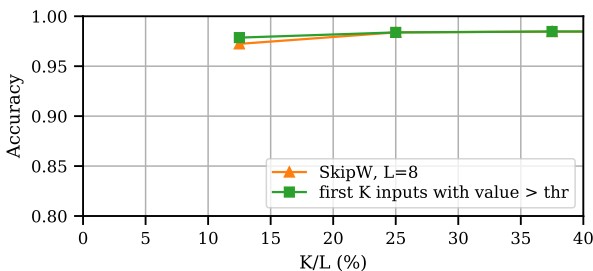

Figure 36: Other variants of $selectK$ do not affect accuracy on HAR with $L$=8.

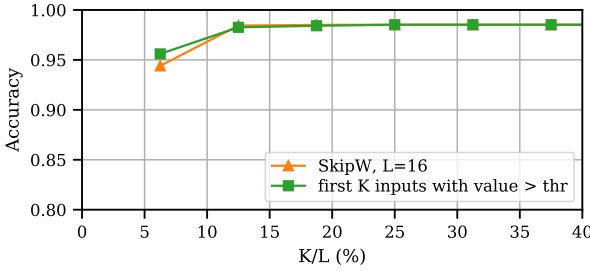

Figure 37: Other variants of $selectK$ do not affect accuracy on HAR with $L$=16.

