# OpenReview forum: "SkipW: Resource Adaptable RNN with Strict Upper Computational Limit"
_ICLR.cc/2021/Conference — ICLR 2021 Poster_

### Official Review · AnonReviewer1 · 2020-10-16
**A simple and effective idea which can have real-world impact - can benefit from stronger experimentation**

**Rating:** 6
**Confidence:** 5

**Review:**

Summary:
The paper proposes Skip-Window or SkipW an abstraction encapsulating RNN cells to actively skip updates similar to some earlier works like Skip-RNN, Skim-RNN, and ThrRNN. The novelty of the method comes is in having control over the total updates to control the overall computational budget compared to previous methods which didn't provide deterministic upper bounds and varied depending on the inputs. The idea is very simple and straightforward and can be looked at as a logical extension to the Skip-RNN line of work combined with a windowed approach on time series as used in ShaRNN (Dennis et al., NeurIPS 2019). The entire time series is divided into Windows of length L (which is a tunable parameter) and each window has a precomputed (from the final hidden state of the previous window) per-time step (update inside the window) importance vector which can be used as an indicator to update or not to update following the binarization as done in previous methods. The strict sparsification of this per-window importance vector to have only K non-zeros per window helps reduce compute to an upper bound ratio of K/L. The method further uses another threshold term over the sparsified importance vector to control finer budget requirements if needed. The experimentation is done on 2 tasks HAR-2D-Pose (with 32 time steps) and Adding task with 50 timesteps. The evaluation shows that Skip-Window shows good performance./accuracy compared to previous flexible RNNs with a reduction in the total number of updates. Finally, the impressive part of the paper is the real-world evaluation on Jetson Nano with a more complex workflow involving pose estimation from images for HAR-2D-Pose.

Pros:
1. Simple and elegant idea. SkipWindow solution is in theory generalizable to multiple RNNs without much hassle. The abstraction could be thought of as an independent outer layer over RNNs similar to most other works in the space.
2. Related work section is very thorough and the claims are grounded.
3. The architecture is easy to understand along with the aspects of training
4. The experimentation on both datasets reveals interesting insights and showcase the advantage of SkipW over other methods in both accuracy and computational budget.
5. I highly commend the deployment experiments and evaluating it with complex workflow and showing how skipping updates and inputs can help compute latency and resources.
6. Plots are very clean and appendix and ablation are good.

Weaknesses:
1. While I feel the idea works well and is elegant. It still is a combination of a couple of known techniques (mentioned in summary) which limits the novelty somewhat. But that doesn't stop the method from being useful.
2. Figure 2 is not required or needs redesigning as the equations were much clearer than the figure and the gates just made it hard to understand.
3. I understand while K/L upper bound is guaranteed, the authors want to use thr for finer control, I just feel, it might not be required for super long sequences. - Just a comment.
4. My major issues are with experiments. While I like the work on HAR-2D-Pose and agree it is a good choice of dataset. I also want to see the success of Skip W on long term dependency tasks. I wouldn't count adding tasks as it can be trivially solved using good initialization (Henaff et al., ICML 2016). While I appreciate the efforts, I would like to see results on at least one or two (at least two preferred but I understand the time limitations, so one good dataset also works for the discussion phase) more real-world datasets to check the generalization. It can be Keyword spotting, phoneme detection, or even noisy-CIFAR. Something around 100 timesteps or more would be great

I am asking the authors to add these things during rebuttal or give reasoning corresponding to it. This is a key question to be addressed during the discussion. Note that I do not need deployment for these datasets but would like to see these numbers in the paper for more datasets to make the paper stronger. The reasoning for the long-range is because the gains would be more profound there than in 32 time steps.

5. While the K/L based on importance makes sense per window. Sometimes, RNNs tend to identify signatures randomly. I suggest the authors add a comparison with random K choice instead of top K choice per window and have multiple variants of it like periodic sampling or all inputs being from start or end (these are deterministic for prediction time). This experiment will help us determine if the importance vector actually contributes to the decision making on skipping updates. The rationale behind this comes from the action recognition literature where people have shown that random sub-sampling works decently well compared to intelligent sub-sampling. Another thing to add here is because the importance vector per window is chosen based on the previous window it might not be optimal and maybe random selection might work fine. If random sampling (even not so random like periodic etc.,) works well, then I am not sure about using importance vectors anymore.

Without these baselines, it is hard to argue otherwise. I strongly suggest authors pursue this to make the claims more solid. This is another thing that needs to be addressed during the discussion.

Decision:
Even though the novelty is slightly limited, I like the idea for its ease, decoupled natured and potential generalizability along with the control on computational budgets. I appreciate the authors for ablation and on-device experiments which are very thorough. The only issue I found with the paper is the experimentation and baselines. I want to see at least one long term dependency task (real-world) and baselines that evaluate if the importance vector is even needed with the simple sampling strategies among each window.

I am very much willing to increase the score based on discussion and the improvements on the experimentation front.
--------------------------------------------------------------------
Edit after rebuttal and discussion.

I thank the authors for extra experimentation to showcase the effectiveness of SkipW. While most reviewers here agree that the novelty is limited (that doesn't stop it from being useful), I strongly think the impact due to SkipW will be translated to the real-world. There has been some discussion on the datasets, which I agree are not extensive making the initial experimentation weak. However, the new experiments compensate to an extent and I would like to recommend a weak acceptance with a score of 6 (I am still between 6 and 7, waiting for other reviewers to pitch in).

---

> ### Author Response · Authors · 2020-11-16
> **We agree. Note that the importance vector is also used to discard useless inputs and save energy.**
>
> Thank you for your review. We were very excited about the results of our approach on real platforms. We are happy you share our enthusiasm!
>
> **Q1.1: While I feel the idea works well and is elegant. It still is a combination of a couple of known techniques (mentioned in summary) which limits the novelty somewhat. But that doesn't stop the method from being useful.**
>
> We think that our approach is not the same as ShaRNN (Dennis et al., NeurIPS 2019). Both use the notions of windows of inputs. However, ShaRNN focuses on parallelizing computation by processing each window independently before aggregating them using a higher level RNN. Our approach computes an importance vector for the inputs in each window but otherwise processes the non-skipped inputs sequentially.
>
> **Q1.2: changing thr might not be required**
>
> As you noted this is not the focus of our work. However we felt that readers may be curious what the impact of changing thr would be. Would you suggest removing it completely?
>
> **Q1.3: more experiments**
>
> We agree, are working on them and hope to have additional results available soon.
>
> **Q1.4: random or deterministic sampling to assess the interest of importance vectors**
>
> Another good point. While additional experiments would be nice indeed, we can already bring some answer to this question and plan to incorporate them in the paper. Campos et al. (2018) have compared their skipRNN approach to random sampling RNNs on multiple data sets. skipRNN significantly outperforms random sampling RNNs. Considering that skipW performs either better than or as well as skipRNN, we believe skipW will outperform random sampling.
> Furthermore, the importance vector in skipW is also used to discard unnecessary inputs and to allow the model to operate below the computational upper bound. This can lead to significant computational savings without impacting accuracy. So this can save a lot of energy as well. Of course a random sampling ratio could be chosen to maintain accuracy. However, as the number and locations of inputs processed in skipRNN and skipW are dynamically chosen based on the sequence processed, we think that random sampling will never reach the same sampling efficiency. This is also supported by the random sampling experiments mentioned in the previous paragraph.
> We would be very happy to hear your thoughts about these two points.

---

> > ### Comment · AnonReviewer1 · 2020-11-16
> > **Thanks for the rebuttal; couple of thoughts**
> >
> > Hi team,
> >
> > Thanks for your response. Let me clarify some things.
> >
> > Q1.1: I didn't say the approach is the same as ShaRNN. I rather said it takes the idea of windowing and combines with the skip-rnn line of work. That doesn't mean the approach is similar but rather a combination of two directions.
> >
> > "The idea is very simple and straightforward and can be looked at as a logical extension to the Skip-RNN line of work combined with a windowed approach on time series as used in ShaRNN (Dennis et al., NeurIPS 2019)." -> This no way means things are similar to ShaRNN.
> >
> > I wanted to clear that if it was confusing to the authors.
> >
> > Q1.2: I don't mind the thr expts, but if you are short on space I would rather push them to the appendix. I think adding more normal experiments should take the front seat and thr can be referenced into an appendix if needed.
> >
> > Q1.3: Looking forward to them.
> >
> > Q1.4: While I agree with the high-level stuff said here, the random sampling in Skip-RNN is going to be very different from the one in SkipW. because of the presence and use of windows for doing the sampling. It becomes local than global, which makes it slightly disconnected from the random sampling of Skip-RNN. I also understand SkipW can throw away inputs, which you can do for random sampling as well. So having random sampling expts per window are going to help make the case stronger. Also, when I mean random-sampling, this also means some variants of structured sampling like periodic, first k, and last k. not just random k.
> >
> > Hope this helps.

---

> > > ### Author Response · Authors · 2020-11-23
> > > **additional baselines and experiments included**
> > >
> > > Thank you for your clarifications.
> > >
> > > **Q1.3: more experiments**
> > >
> > > We have included new experiments on sequential MNIST and IMDB, a a NLP tasks. We unfortunately lacked time to use one of the data sets you suggested. While these two tasks were not part of your suggestions, we chose them for two reasons. First, two reviewers requested each of these two experiments. Then, theses tasks were analyzed in the SkipRNN paper. This gives us a point of comparison, a sanity check to verify our results and make sure we are optimizing hyper parameters reasonably well for both SkipRNN and SkipW. We have achieved better tradeoffs for SkipRNN in our experiments than in the original paper. We are therefore confident that the comparison is fair. In addition, in both tasks sequences are much longer than 100: 781 for MNIST and 200 for imdb. SkipW achieves tradeoffs similar to SkipRNN on MNIST, and slightly worse on IMDB. MNIST also provides another interesting comparison between the skip patterns learned by both methods. We also included a comparison to recent feature selection results (for MNIST), showing that SkipW achieves result close to some of them.
> > >
> > > **Q1.4: random or deterministic sampling to assess the interest of importance vectors**
> > > On that front, we are happy to report that we have included random and periodic baselines in the main experimental section, and several additional baselines in Appendix H. We have performed three sets of
> > > experiments on HAR to study the sampling mechanism:
> > > - a comparison to random sampling, periodic sampling and sampling the first K elements of a window. SkipW is better than these approaches.
> > > - a comparison to alternative sampling mechanisms: rather than a weight vector, we tried random sampling and using the first or last elements of a window. SkipW is also better most of the time, showing the interest of importance vectors.
> > > - replacing TopK by another select_K mechanism. Here we only tried using the first elements of the window (above the threshold). The performance is similar to TopK.
> > >
> > > We believe these experiments make the paper (even) more interesting. We hope you will share our sentiment!

---

> > > > ### Comment · AnonReviewer1 · 2020-11-24
> > > > **Appreciate the extra experiments; couple of questions**
> > > >
> > > > The authors did a good job in adding new experiments to make the paper stronger including sequential MNIST and IMDB, along with visualizations, and multiple sampling strategies as well. I have a couple of questions.
> > > >
> > > > I will go sequentially.
> > > >
> > > > 1) For Figures 3 and 4, when you do random sampling, it is not mentioned how many times each experiment was run. The mean and variance are important for uniform random sampling experiments. Please add this in the next revision, else it might seem like cherry-picking. Other strategies like periodic or first K don't have this variance associated with them.
> > > >
> > > > 2) For Figures 5 and 6, the trade-off SkipRNN plot doesn't go to the ranges where SkipW is also being presented. At the high compute arena they look close, so I strongly suggest the authors pull the tradeoff line to wherever SkipW is working or give reasoning on why SkipRNN might fail there.
> > > >
> > > > 3) For the sequential MNIST experiments, there are so many moving parts and I can't compare apples to apples for SkipW vs SkipRNN for a given accuracy of % of inputs processed. Why is it the case? Can the authors shine some light on what is happening here? SkipRNN has 96.1% and 89.7% accuracy at 15.6% and 9.5% inputs processed and the closest I can compare it to SkipW is 96.4% for 26.8% inputs process and I have no datapoint for the 10% input range or 90% accuracy. I would appreciate an explanation for this (Table 3). This is the rationale behind more datasets, to figure out the generalizability and adaptability.
> > > >
> > > > 4) This is a comment, the fact that only 3 pixels per line are needed in MNIST might suggest it is not a great dataset for benchmarking. I understand the constraints and hence have no request from the authors on this front. I also note that the 3 pixels are selected adaptively, so random sampling might not work, but I would love to see random sampling work here. I am sure periodic will do great due to the cleanness of the MNIST dataset (centered and not many feasible pixel values). Your illustrations can also be thought of as attention which also provides very similar results for input ommision.
> > > >
> > > > 5) I like the random/periodic sampling experiments, Figs 32-33. They confirm what I was pointing about. The accuracy gains vs the compute cost involved for computing the window importance vectors is a debate that needs to be addressed. I don't see a periodic sampling in these two figs but if period and first/last K have minimal accuracy drop compared to SkipW during prediction, what is the extra FLOPs (compute) overhead of SkipW for the ~1% gains at low-input processed regime
> > > >
> > > > I like the idea a lot, but given the idea is tackling a setting where such heuristics are cheap, I keep asking the authors for supportive arguments for SkipW. I am still in support of the paper, but I need answers for the new questions to be able to stand-up for it.

---

> > > > > ### Author Response · Authors · 2020-11-24
> > > > > **couple of answers**
> > > > >
> > > > > 1. Done
> > > > > 2. We have added some points to the IMDB figure. For MNIST, we have not been able to produce better SkipRNN models. Unfortunately, at some point both SkipW and SkipRNN stop learning and basically skip all inputs. We have added some failure examples in Table 3.
> > > > > 3. The number of inputs processed is only controlled indirectly through $\lambda$ for both methods. This makes it difficult to train methods processing a similar number of inputs. Please do note that the table contains fewer data points than the figure. We now realize this is misleading and will correct that. In particular, there is a SkipW model achieving 96.01% accuracy and 15.07% input processed, which is really close to the SkipRNN model at 96.1% and 15.6%
> > > > > 4. Note that we provide a non-sequential random sampling baseline for MNIST (71% accuracy at 3.2% inputs).
> > > > > 5. Thank you! SkipW retains a significant computational advantage over the variants not using the importance vectors, because SkipW can process fewer than K inputs in every window. We have included two additional figures (34 and 35) using the actual number of inputs processed rather than K/L. While not as good to compare accuracy, we new realize they are needed to correctly reflect the tradeoffs. We expect this advantage to be even bigger on MNIST and adding task, as there are windows without any information in these tasks.

---

> > > > > > ### Comment · AnonReviewer1 · 2020-11-24
> > > > > > **Thanks**
> > > > > >
> > > > > > Thanks a lot for the answers. I will increase my score to an accept after discussing with other reviewers and AC :)
> > > > > >
> > > > > > I hope you see how these experiments made the paper better.

---

### Official Review · AnonReviewer4 · 2020-10-23
**Interesting work**

**Rating:** 6
**Confidence:** 2

**Review:**


This work introduces Skip-Window (SkipW), an approach that allows RNNS to have improved computational efficiency at the cost of accuracy. SkipW adds a procedure to existing RNN cells that allows them to process fewer inputs while remaining in a strict computational budget. This work demonstrates the benefits of SkipW through experiments on multiple data sets.

This work proposes a structured procedure to process fewer inputs during inference while abiding by a computational budget. Each skip seems to be calculated over a window of inputs in a sequence, thereby minimizing inter and intra sequence variability in computation. This work does seem to have merit in a practical setting where the availability of computational resources can vary. The ability to manage dynamic budget, though not totally novel (ThrRNN also has the thr param), does provide an advantage with SkipW being perhaps more flexible with both K and thr usage.

This work empirically demonstrates the benefit of SkipW  over several baselines on two data sets. There are, however, some points that need to be addressed.





Figure 2 and equations 5-10 seem to be a bit unclear. Figure 1 and the first two paragraphs of section 3 seem to imply that  \hat{u}_{W} is calculated once as an L length vector at the beginning of an L sized window.   Here \hat{u}_{W}^{K}  are the selected K entries for the whole L sized window. Therefore what does \hat{u}_{W,t}^{K}  denote.

In Figures 3 and 4 what is input processed % ? Is it K/L. For example, in Figure 3 and in the case of L=8, the setup K=1 implies K/L = 0.125 , but the K=1 point for SkipW is at less than 10% mark on the xaxis. Furthermore, how is it that K=[3-8] all have roughly the same input processed % ? Its the same case for K=[1-4] and K=[3-16]


In Figure 4, the number of points for ThrRNN seem to be much more than SkipW. This seems to imply much more finer control over computational cost for ThrRNN as compared to SkipW (more coarser control). What are the authors' thoughts about this?


Figure 6 seems to imply that the inputs in p3 and p4 can be more easily dropped as compared to p1 and p2. What are the  authors' thoughts about the connection of this ease of dropping inputs to the various attention mechanisms that are available in literature. Furthermore, is it the case that later inputs will typically always have more tendency to be dropped in SkipW ?


Figure 7 seems to be a bit unclear. Is the accuracy being depicted the relative accuracy as compared to thr=0.4 and K=8? Furthermore, how is the accuracy being calculated as the thr,K are changing on the fly as time/sequences go on? This might imply that different thr,K values see different data? Is the same exact sample being used to calculate the accuracy?


It seems that L-K denotes the minimum number of inputs that will be skipped (as there is a further 'binarize' that runs on the K inputs). Therefore, it seems that the system will under utilize the available computational capacity. This might not be desirable as any loss in accuracy might have been reduced if all available computational capacity was used. What are the authors' thoughts about this?


Perhaps a text processing task in NLP would have made the results stronger as in practical scenarios, this is one of the common modalities where RNNs are used.

---

> ### Author Response · Authors · 2020-11-16
> **Here are some clarifications**
>
> Thank you for your review.
>
> **Q4.1: Figure 2 and equations 5-10 seem to be a bit unclear. What does \hat{u}_{W,t}^{K} denote.**
>
> We think there no \hat notation in our paper, so we will assume you meant \tilde. If not, please do correct us.
> \tilde{u}{W} is indeed computed once at the beginning of an L sized window. But in order to change the trade off during the processing of this same window, this vector is available at every time step and denoted by \tilde{u}{W,t}, where we refer to the time step rather than the window index as in figure 1. We thought this was clearer, but we will clarify.
> Therefore at each time step t, \tilde{u}_{W,t}^{K} is the current L-vector values which will depend on the current threshold and the current K parameter.
>
> If the threshold and K parameter are considered fixed for a window duration, the vector \tilde{u}{W} can be topked and fbinarized and therefore \tilde{u}_{W,t}^{K} is necessary.
>
> **Q4.2: In Figures 3 and 4 what is input processed % ?**
>
> *input processed %* in figure 3 and 4 refers to the percentage of the inputs that the RNN models do not skip and process. This is a measure of the computational cost of the model.
>
> **Q4.3: For example, in Figure 3 and in the case of L=8, the setup K=1 implies K/L = 0.125 , but the K=1 point for SkipW is at less than 10% mark on the xaxis.**
>
> You are right. K/L is an upper bond on the number of inputs processed. But unimportant inputs are also discarded by the f_binarize function. So the actual number of inputs processed can be lower than K/L.
>
> **Q4.4: Furthermore, how is it that K=[3-8] all have roughly the same input processed % ? Its the same case for K=[1-4] and K=[3-16]**
> The number of inputs processed is low to start with because of the f_binarize function. As there are already few inputs processed with K=L, lowering K has not much impact in this case as the upper bound of K/L is already larger than the average input processed with K=L.
>
> **Q4.5: In Figure 4, the number of points for ThrRNN seem to be much more than SkipW.**
>
>  Indeed, ThrRNN adapts using the threshold (which can take any value), while the K parameter of skipW can only take integer values in {1,..,L}. However, as the upper bound is K/L, only K enables to match a constraint. thrRNN therefore does not allow to strictly limit computation. Using the threshold can also be done in skipW. This allows finer control but brings no additional benefit with respect to the constraint. You can see the impact of changing thr in skipW in figure 9 in the appendix, where modifying thr provides a similar  level of control as in thrRNN, but from more trade-off points corresponding to the different K values.
>
> **Q4.6: Figure 6 seems to imply that the inputs in p3 and p4 can be more easily dropped as compared to p1 and p2.**
>
> We are not sure we understand the question, so please follow up if we miss the mark. In this figure, inputs in p_1 and p_3 are completely analyzed for high K and increasingly skipped when K decreases. About half the inputs in p_4 are analyzed for high K, and increasingly skipped when K decreases. Most inputs in p_2 are skipped, not matter the value of K. We believe this behavior is due to the position of the markers (in p_1 and p_3) and does not reflect a tendency of SkipW.
>
>
> **Q4.7: Figure 7 seems to be a bit unclear.**
>
> The accuracy and average number of inputs processed that are reported are average values on a set of 5751 test sequences. Figure 7 analyzes the impact of changing thr and k inside a sequence, so on the fly as inputs go on. For example, the points (accuracy and computational cost) in the figure where x axis = 50% correspond to changing from (thr=0,4;K=8) to (thr=0,65;K=1) in the middle of every sequence out of 5751, so after seeing or skipping half of the inputs. In other words, the first half of each sequence is processed using (thr=0,4;K=8) and the second half using (thr=0,65;K=1). A prediction is only computed at the end of the sequence. This is true for every sequence. Values plotted at x axis = 30% corresponds to changing the thr and K after one third of every sequence and so on.
>
> **Q4.8: It seems that the system will under utilize the available computational capacity. This might not be desirable as any loss in accuracy might have been reduced if all available computational capacity was used.**
>
> Indeed, the system may under utilize the available computational capacity, but this should have little or no impact on accuracy. While we do want skipW to be flexible and be able to operate under a strict computational constraint, we also want to train a model that will use as little inputs as possible, while maintaining accuracy. This saves energy and we think there is no point in processing inputs if it is not useful. This is related to your question 4.4: for high values of K (figure 3 and 5), the same number of inputs are skipped so as not to waste resources.

---

### Official Review · AnonReviewer2 · 2020-10-26

**Rating:** 5
**Confidence:** 5

**Review:**

This submission presents an extension of SkipRNN, Skip-Window, that splits input sequences into windows of length L from which only K samples can be used. This guarantees that the computational budget is never exceeded. Skip-Window implemented this inductive bias by predicting L updating probabilities in parallel at the beginning of each window. L needs to be set prior to training, whereas K can be modified at test time. The model is evaluated in two tasks, namely a synthetic adding task and human activity recognition. Authors report latency and energy consumption in small platforms, showing the impact of this research direction in real applications.

My main concern regarding the proposed architecture is that it limits the types of skipping patterns that can be discovered -- whereas the original SkipRNN can in principle learn any such pattern. For instance, Skip-Window cannot produce the skipping patterns shown in Figure 6 in Campos et al. (2018) unless K=L or L=1 (in both cases, Skip-Window reduces to SkipRNN). This can even be seen in the results for the adding task, where Skip-Window is actually *not* solving the task for most values of K (c.f. Figure 4). Recall that the output distribution has a variance of 0.166, and Campos et al. define solving the task as achieving an MSE two orders of magnitude below such variance. The reason for this is that Skip-Window with 5<K<L will miss the second marker in some sequences, as it needs to guess its position -- which is random within the second half of the sequence. For K<5, there’s a chance that Skip-Window will miss the first marker as well. As a sanity check, I suggest that authors report the percentage of first and second markers that are missed in the adding task as a function of K. Plotting the MSE of an LSTM or GRU that skips inputs randomly, and varying the fraction of skipped inputs as in Campos et al., would also provide context for the presented error rates.

Despite the imposed constraints in terms of skipping patterns, Skip-Window can be useful in tasks where the input signal can be downsampled more uniformly. After all, the adding task is a challenging problem for RNNs that skip input samples as missing one of the markers will massively increase the error rate. This potential is shown in the human activity recognition (HAR) results. However, I believe that more experimental evaluation is needed before this paper can be published at ICLR. First, the input sequences for HAR are extremely short (32 timesteps), which makes it difficult to draw strong conclusions. Second, since the Skip-Window architecture limits the types of skipping patterns that the model can discover, it is difficult to claim that Skip-Window is a generic RNN architecture unless experiments on more domains are reported (e.g. sequential MNIST, NLP).

---

> ### Author Response · Authors · 2020-11-16
> **Main concern is not correct: skipW can discover any skip pattern and restrict patterns to match constraint**
>
> Thank you for your review.
>
> **Q2.1: My main concern regarding the proposed architecture is that it limits the types of skipping patterns that can be discovered -- whereas the original SkipRNN can in principle learn any such pattern**
> We believe this statement and several elements in the associated paragraph are not correct. We argue and would like to discuss that the following statement is more accurate.
> *Both the proposed architecture and skipRNN can in principle learn any skipping pattern. Unlike skipRNN, the proposed architecture can, during inference, forbid some skipping patterns to avoid exceeding a computational limit. In practice, both the proposed architecture and skipRNN have some limits on the skipping patterns that can be learned.*
>
> Our architecture can be used with any K. Our architecture can be used with K=L and efficiently skip inputs. There is therefore no restriction on the types of skipping patterns learned and used by our architecture, as in skipRNN. In theory, SkipW can solve any task. This can for example be seen in Figure 4, where skipW solves adding task for K=10=L.
> On a more technical side, K=L or L=1 are not the only configurations allowing skipW to reproduce the skip patterns of figure 6 in Campos et al. (2018). The longest sequence without skip in that figure seems to be of length 14. In theory, SkipW can achieve such skipping patterns for any K>=7 and L >= K, by examining the last K inputs of one window and the first K inputs of the following window. That being said we acknowledge that for any K<L, a high enough frequency will be problematic.
>
> By lowering K, skipW can be used to match a computational constraint by forbidding some skip patterns. If the optimal skip pattern is forbidden, skipW can in theory fall back to another good skip pattern. In other words, skipW prevents some patterns only when computational limitations would not allow such patterns to be analyzed. Our experiments suggest it works in practice. Sometimes there is no impact on accuracy (figure 5, K changing from 5 to 2), sometimes there is (figure 5, K changing from 2 to 1). Whether the decrease in accuracy is acceptable or not depends on the task. However, without skipW, there is no possibility to match the constraint. This is the real improvement over skipRNN. For any non-trivial task, we agree that, with L large enough and K small enough, skipW will fail.
>
> But this is no the only improvement: skipW can also achieve better accuracy / computational trade-off than skipRNN. When K=L, skipW does not reduce to skipRNN. The skip mechanisms are different, and lead to different trade-offs. In practice, skipRNN seems to be more limited than skipW in the skip patterns it can learn. For example, skipRNN is unable to skip long sequences of inputs. This is visible in our experiments but also in Campos et al. (2018). For example, in figure 6 of Campos et al. (2018), skipRNN samples multiple inputs at the beginning as you pointed out but also samples individual inputs several times throughout the sequence even though these inputs bring no information because of aliasing. This is also visible in other figures. Because skipRNN is dragged down by its inability to completely stop sampling where there is no information, when trying to further reduce the number of inputs processed by increasing lambda, skipRNN stops analyzing interesting inputs. We suspect that skipW has some limitations as well, but in our experiments, skipW achieves comparable or better results in terms of accuracy and computational cost trade offs than skipRNN. So we argue that the potential limitations of skipW are in practice less penalizing than or at least comparable to the limitation of skipRNN.
>
> We think that the points above could be emphasized better in the paper. We plan to incorporate them. Thank you for provoking this discussion.
>
> We also agree that additional experiments would be a great addition to the paper. We are working on them at the moment and hope to fulfill some of your suggestions before the end of the discussion period.
>
> Adding a baseline randomly skipping inputs in adding task is however easy, as Campos et al. (2018) provide these results. The task fails almost immediately, even when skipping only 2% of inputs. For some perspective on other data sets, please see our answer to Q1.4 of reviewer 1.

---

> > ### Comment · AnonReviewer2 · 2020-11-17
> > **Skipping patterns & Solving the adding task**
> >
> > Thanks for your reply. I'm looking forward to seeing results on other tasks.
> >
> > I agree that there was a mistake in my original review: when $K=L$ and $L>1$, SkipW is not exactly the same as SkipRNN -- as it will decide whether to update or copy the next L inputs in parallel instead of one by one. As authors pointed out, this might help skipping large subsequences, something that SkipRNN struggles with due to the additive nature of its updating probability.
> >
> > Let me clarify my original comment on the capability of SkipW for discovering the same patterns as SkipRNN. I didn't mean that there does not exist a hyperparameter combination that enables reproducing the same behavior as in Figure 6 by Campos et al. (2018). What I meant is that SkipW (K<L) implements a particular inductive bias towards "periodic" sampling, as at most K samples per window of size L can be used. As an example, let's assume we would like to use at most 10% of the samples and thus set $K/L=0.1$.  If the optimal subsampling scheme consists in sampling the first 10% elements of the full sequence, SkipW will not be able to discover such scheme -- it is forced to use at most 10% of the samples per window (not per sequence). This inductive bias might be good enough for many tasks, but I believe that it's limitations should be highlighted in the manuscript too.
> >
> > I would like to stress one of my main concerns, which has not been addressed in the comment: SkipW fails at the adding task for most values of K. Note that a similar concerns was raised by AnonReviewer3. Plotting the MSE is misleading, as the task is not really being solved (which is defined as obtaining an MSE below 0.0016). I suggested to include the random skipping baseline in Figure 4, for different values of the skipping probability, so that we can measure whether SkipW is obtaining any benefits with respect to random skipping when K<L -- after all, SkipW is missing markers. This is a similar methodology to that in Campos et al. (2018), who include a random skipping baseline per value of $\lambda$ in SkipRNN. In the case of SkipW, there should be one baseline per value of K; some values of K can be skipped to reduce the number of models that need to be trained, although note that models converge in a few minutes in this task.

---

> > > ### Author Response · Authors · 2020-11-19
> > > **Skipping patterns & random sampling and markers missed on adding task**
> > >
> > > Thank you for your answer. We agree that for a given K, SkipW limits the number of updates per window and that skipRNN does not have such a limitation. We will state this more explicitly. However, such a constraint is mandatory to satisfy computational constraints on portions of the sequence, including on the whole sequence. So we argue that this limitation is one reason why SkipW is interesting. In addition, we would like to raise two points:
> > > - In our opinion, the example in the answer and in particular the sentence "SkipW will not be able to discover such scheme" do not represent how SkipW works. To make sure there is no misunderstanding, we would like to clarify. As you mentioned in the summary, K is selected at test time.
> > > 	- During training, there is no restriction on the value of K and no restriction on the patterns SkipW can explore and therefore it can discover any skipping scheme. In your example, we expect that SkipW will correctly identify that it must sample the first 10% of the sequence. This example actually corresponds to the first marker of adding task. Figure 13 in the appendix illustrates that SkipW has correctly learned that it must sample the first 10% of the sequence and then stop (until the second marker can appear). SkipRNN has not learned it must stop (SkipRNN is labeled as ThrRNN - ThrRNN with a constant $thr=0.5$ is SkipRNN. Also something we will make clearer).
> > > 	- During inference, K is selected to fit a computational constraint. That is a system / user choice. In practice, this restricts the type of skip patterns used while the constraint is enforced. But again, this has no influence on training. In your example, SkipW would indeed stop sampling some inputs in the first 10% of the sequence if a strict constraint is applied at the beginning. If no strict constraint is applied (K=L) or if the constraint does not forbid it, SkipW would sample the first 10% of the sequence and then stop. This can be observed for adding task in figures 6 and 15.
> > > - We would like to stress that, despite this limitation, we achieve better or comparable tradeoffs than/to skipRNN.
> > >
> > > A comparison to random sampling is indeed a good idea. We also agree that training is fast for adding task, but our time and computational power is limited. At the moment we focus on new datasets, as this has been requested by all reviewers and we expect this to be the most time consuming task. We apologize that this delays our answers somewhat. As an alternative to empirical random sampling on adding task, we propose to use the theoretical best average error such a strategy can achieve. As you pointed out, the variance of the task is 0.166 as the variance of each value is 1/12. A model performing perfect summation of the values that are not skipped will therefore achieve an average error of P(skip) * 1/12 * number of markers. We have updated Figure 4 of our paper with this metric. Both skipRNN and SkipW is well below this limit. We have also added the proportion of markers skipped (Figure 12).

---

> > > > ### Comment · AnonReviewer2 · 2020-11-19
> > > > **Skipping patterns & random sampling and markers missed on adding task**
> > > >
> > > > Thanks for your reply. I appreciate the effort in clarifying how SkipW learns to skip patterns.
> > > >
> > > > I believe that Figure 12 in the Appendix proves that SkipW misses markers in the adding task for $K<L$. If the goal were to solve this task, one would need to rely on the skipping patterns discovered by SkipW, without forcing strict computation limits at test time (i.e. $K=L$). In this case, the distribution of marker positions will determine whether SkipRNN or SkipW are better suited for the task -- after all, it comes down to a matter of choosing the right inductive bias for the skipping pattern. Figure 13 shows that SkipRNN is able to skip inputs after finding a marker, but it is unable to skip long subsequences due to the additive nature of the updating probability. On the other hand, SkipW always uses the first 5 elements in the sequence (i.e. it is unable to skip inputs 2 to 4 if the marker is in the first position). In some sense, it is relying on a prior learned during training (and not so much on a reactive skipping pattern). What I mean by this is that the choice between SkipRNN or SkipW ($K=L$) will depend on the task, as well as on a proper choice of $L$.
> > > >
> > > > I understand that we have suggested multiple experiments and it is difficult to run them all in such a limited time. I agree that reporting results on other tasks/datasets is more important than a deeper analysis on this synthetic task, and appreciate the effort made to provide more results. As an alternative, if authors have the checkpoints for those models, I would suggest reporting whether the adding task is solved (Yes/No) for the different settings in Figure 12.
> > > >
> > > > I am looking forward to seeing results on other tasks.

---

> > > > > ### Author Response · Authors · 2020-11-23
> > > > > **discussion incorporated in paper and requested experimental results in paper**
> > > > >
> > > > > We think we are in agreement about the merit of each method, although we would like to correct that SkipW also dynamically react to the inputs, but slower than skipRNN. On adding task (Figure 15), after the second marker has been detected on adding task, SkipW stops sampling inputs. But this only starts at the beginning of the next window whereas skipRNN does it immediately, a drawback of SkipW for sure, though SkipW is still better.
> > > > >
> > > > > We think this discussion has been very interesting and raised several good points. We have decided to incorporate the essence of it in the paper. We hope you are fine with this.
> > > > > Specifically, the main experimental section of the paper now mentions adding task failures and we have added some perspectives to the discussion on skip patterns.
> > > > > In the appendix (C.3), on adding task, as it is particularly well suited to compare skip patterns, we have tried to summarize the present discussion. Please let us know what you think and whether something is missing.
> > > > >
> > > > > We have added a random sampling baseline on Adding Task and HAR. In both tasks, both SkipW and SkipRNN beat these baselines.
> > > > > In addition, we have included a discussion of failures on adding task and additional experiment on sequential MNIST and imdb, a NLP task.

---

> > > > > > ### Comment · AnonReviewer2 · 2020-11-24
> > > > > > **Thanks for the updates**
> > > > > >
> > > > > > Thanks a lot for including new experiments, visualizations and the main points of this discussion. The updated manuscript provides much more insight about the proposed method and how it compares to existing work. I will reconsider my original recommendation once I go through the new content.

---

### Official Review · AnonReviewer3 · 2020-10-28
**Interesting paper, but lack of motivation**

**Rating:** 6
**Confidence:** 3

**Review:**

The authors proposes a new framework skipW to strictly limit the computation in RNN.

Pros:
1. The idea of strictly limiting the computation in RNN is new.
2. The summary of related works is clear.

Cons:
1. The motivation why the authors wants to enforces the strict constraint on the number of updates is unclear:
a. What if a consequent subsequence in a sequence is important? Then limiting processing only the K of L elements will omit this subsequence.
b. Playing with lambda in equation (11) may also give you a tradeoff in computation and model performance and it will not omit the consequent subsequence example I give in (a).
2. Is the training harder if we use less inputs and use error gradients than the training with the whole sequence? Although the computation time is less for a forward process, the training may be harder and take more time. This detail needs to be included.
3. ThrRNN did one experiment in MNIST. I would like to see the comparison on the MNIST dataset.

Clarity:
1. The paper is clearly written in general.

Originality:
1. As far as I know, the paper is novel.

Minor:
1. It somehow sounds mysterious to me that the SKipW model can learn the adding task. The $\tilde{u}_{W,t+1}$ only takes inputs at the beginning of the length L subsequence. It may not see the inputs where the marker is. Then how can the model learn to not to skip the inputs where the marker is?

---

> ### Author Response · Authors · 2020-11-16
> **Strict constraints allow the model to run on limited and shared computational resources**
>
> Thank you for your review. We would like to address some of your concerns.
>
> **Q3.1: The motivation why the authors wants to enforces the strict constraint on the number of updates is unclear: a. What if a consequent subsequence in a sequence is important? Then limiting processing only the K of L elements will omit this subsequence. b. Playing with lambda in equation (11) may also give you a tradeoff in computation and model performance and it will not omit the consequent subsequence example I give in (a).**
> a) You are correct: the model might omit some important inputs if more than K of these are present in a subsequence of L inputs. Whether the decrease in accuracy is acceptable or not depends on the task. However, without skipW, there is no possibility to match the constraint and we argue that lowering accuracy is often better than canceling the task. Please also see our answer to reviewer 2. Regarding (b): Playing with lambda allows to obtain an offline trade off that will be fixed. If we understood your comment correctly, this is similar to skipRNN (although the skip mechanisms are different). While skipRNN works well, we would like to stress that, in our experiments,  skipRNN is actually worse than or comparable to skipW. We believe skipRNN has trouble skipping a large number of inputs and cannot properly exploit the structure of the problem, as shown in the detailed experiments on adding task. So we argue that limiting the ability to analyze all elements is at least as good as skipRNN inability to skip a large number of inputs. Furthermore, unlike skipRNN, skipW allows the model to adapt to changing constraints on devices with limited resources, which is the main motivation of our work.
>
> **Q3.2: Is the training harder if we use less inputs and use error gradients than the training with the whole sequence?**
>  During training, to process inputs in batch, we use multiplication by u_t and (1 - u_t) rather than actually skipping inputs. This allows the use of matrix operations on  mini-batches of inputs, which makes training faster than processing sequences one by one and skipping inputs. We will add these details.
>
> **Q3.3: It sounds mysterious to me that the SKipW model can learn the adding task**
> While training, the K parameter can be set to L, therefore the model is able - in regard of the lambda parameter - to use as much inputs as it likes in order to minimize the loss. While training, it learns to minimize the loss, to skip inputs in the irrelevant part of the sequence and spot the markers. More precisely, skipW can be used with any K. When K<L, some skip patterns are forbidden and the model falls back to other skip pattern.

---

### Author Response · Authors · 2020-11-23
**Thank you for your reviews**

Dear reviewers, thank you again for the discussion. We have added additional experiments. Please let us know whether these address your concerns.

Changelog:
- Better highlighting that SkipW is interesting both for the tradeoffs achieved and the bounded computational cost
- Random selection baseline for Adding task (requested by R2)
- Random and deterministic baselines for HAR (requested by R2 and R1)
- Convergence speed comparison (requested by R3)
- Additional experiments (requested by all):
   - sequential MNIST (length 784, requested by R3 and R2)
   - IMDB (length 200, NLP requested by R2 and R4)
- Additional baselines:
   - random sampling baselines on HAR and adding task (R3, R2 and R1)
   - variations of the sampling mechanism (R1)

---

### Decision · Program_Chairs · 2021-01-07
**Final Decision**

**Decision:**

Accept (Poster)

**Comment:**

The authors did a good job responding to reviewer concerns.   While the reviewers still consider the method described in the paper to not be especially novel, at least one is impressed by the practicality.  imo the authors' attention detailed ablations and analysis post-review makes the paper worth including in the conference.